# Admixture has obscured signals of historical hard sweeps in humans

Yassine Souilmi ●[1,14] ✉, Raymond Tobler ●[1,2,14] ✉, Angad Johar ●[1,3,14] ✉, Matthew Williams[1], Shane T. Grey[4,5], Joshua Schmidt ●[1], João C. Teixeira ●[1], Adam Rohrlach ●[6,7], Jonathan Tuke ●[6,8], Olivia Johnson[1], Graham Gower ●[1], Chris Turney ●[9], Murray Cox[10], Alan Cooper[11,12,14] ✉ and Christian D. Huber ●[1,13,14] ✉

The role of natural selection in shaping biological diversity is an area of intense interest in modern biology. To date, studies of positive selection have primarily relied on genomic datasets from contemporary populations, which are susceptible to confounding factors associated with complex and often unknown aspects of population history. In particular, admixture between diverged populations can distort or hide prior selection events in modern genomes, though this process is not explicitly accounted for in most selection studies despite its apparent ubiquity in humans and other species. Through analyses of ancient and modern human genomes, we show that previously reported Holocene-era admixture has masked more than 50 historic hard sweeps in modern European genomes. Our results imply that this canonical mode of selection has probably been underappreciated in the evolutionary history of humans and suggest that our current understanding of the tempo and mode of selection in natural populations may be inaccurate.

The rapidly growing availability of genomic datasets provides a powerful resource that can be used to address a fundamental question in evolutionary biology, namely, the role of natural selection in shaping biological diversity[1]. The proliferation of genomic datasets has been accompanied by parallel developments in statistical methods for uncovering genetic signals of positive selection; however, although the power and precision of these statistical methods has continued to improve, they can be confounded by complex aspects of population history that are not modelled or remain unknown[2]. In particular, previous genomic studies of positive selection typically do not account for past phases of interpopulation mixing (that is, admixture), which can alter genomic signatures of positive selection and either mask these signals or lead to erroneous inferences about the underlying modes of selection[3–6].

In the case of humans, a consistent finding has been the apparent paucity of classical 'hard sweep' signals in modern genomic datasets (that is, where a new beneficial mutation increases to 100% frequency

[1]Australian Centre for Ancient DNA, The University of Adelaide, Adelaide, South Australia, Australia. [2]Evolution of Cultural Diversity Initiative, Australian National University, Canberra, Australian Capital Territory, Australia. [3]Department of Cardiovascular Diseases, Mayo Clinic, Rochester, MN, USA. [4]Transplantation Immunology Group, Immunology Division, Garvan Institute of Medical Research, Darlinghurst, New South Wales, Australia. [5]St Vincent's Clinical School, Faculty of Medicine, UNSW, Darlinghurst, New South Wales, Australia. [6]ARC Centre of Excellence for Mathematical and Statistical Frontiers, The University of Adelaide, Adelaide, South Australia, Australia. [7]Department of Archaeogenetics, Max Planck Institute for the Science of Human History, Jena, Germany. [8]School of Mathematical Sciences, The University of Adelaide, Adelaide, South Australia, Australia. [9]Chronos 14Carbon-Cycle Facility and Earth and Sustainability Science Research Centre, University of New South Wales, Sydney, New South Wales, Australia. [10]Statistics and Bioinformatics Group, School of Fundamental Sciences, Massey University, Palmerston North, New Zealand. [11]South Australian Museum, Adelaide, South Australia, Australia. [12]BlueSky Genetics, Ashton, South Australia, Australia. [13]Department of Biology, Penn State University, University Park, PA, USA. [14]These authors contributed equally: Yassine Souilmi, Raymond Tobler, Angad Johar, Alan Cooper, Christian D. Huber. ✉e-mail: yassine.souilmi@adelaide.edu.au; raymond.tobler@adelaide.edu.au; johar.angad@mayo.edu; alanjcooper42@gmail.com; cdh5313@psu.edu

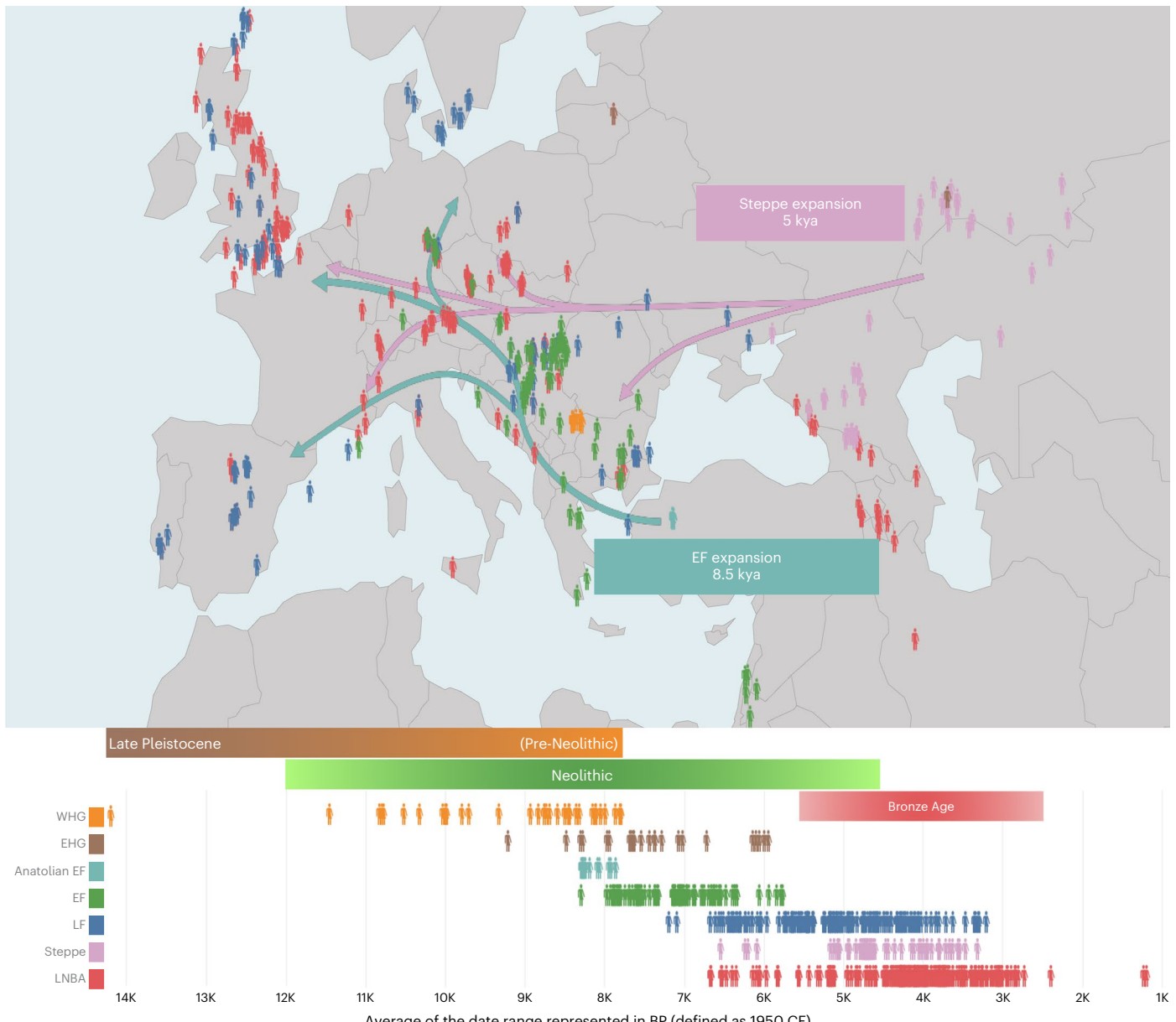

**Fig. 1 | Geographic and temporal distribution of 1,162 ancient Eurasian samples used in this study.** Each human symbol represents a sample, and the colours indicate different populations classified into broad groupings according to archaeological records of material culture and lifestyle (colours indicated at the bottom left-hand side; Supplementary Text 1). Sample ages are represented in the bottom panel in thousand years before present (BP). The green lines depict the generalized migration route of Anatolian EF into Europe ~8.5 ka, where they mixed with WHG (EHG refers to the contemporaneous Eastern Hunter-Gatherers) to create the European Early Farmers (EF). Similarly, the pink arrows represent the generalized movement of Steppe Pastoralists (Steppe; samples east of the Ural Mountains not shown), which resulted in admixture with LF ~5 ka, giving rise to LNBA societies.

in a population). This has resulted in suggestions that humans adapted to new environmental and sociocultural pressures through alternate modes of selection—for example, 'soft sweeps' (where the beneficial allele sits on multiple genetic backgrounds)[7,8] or polygenic selection (subtle frequency shifts across numerous loci with small fitness effects)[9,10]—though the evidence for these processes has also been challenged[5,11–13]. The apparent ubiquity of admixture in human population history[14–16] suggests instead that the absence of hard sweep signals in modern human genomes may be a consequence of the masking effects of historical admixture events. Moreover, growing evidence suggests that admixture events pervade the history of most natural populations[17], which may have led to a potentially biased view of the modes of selection operating in nature[3–5,18].

## Ancient human genomes uncover historical hard sweeps

The recent emergence of population-scale ancient human genomic datasets from Eurasia provides a new means of searching for any hard sweeps that have occurred in the ancestors of modern Europeans, by applying selection scans to genomic datasets that predate known admixture events. Modern Europeans are largely composed of three distinct ancestries—that is, Western Hunter-Gatherers (WHG), Anatolian Early Farmers (Anatolian EF) and Steppe Pastoralists (Supplementary Text 1)—as a result of extensive admixture between these ancestral populations and their descendants that occurred from the Early Holocene to Bronze Age (~12–5 ka)[19–21]. Accordingly, we reprocessed 1,162 ancient western Eurasian genomes (mostly high-density

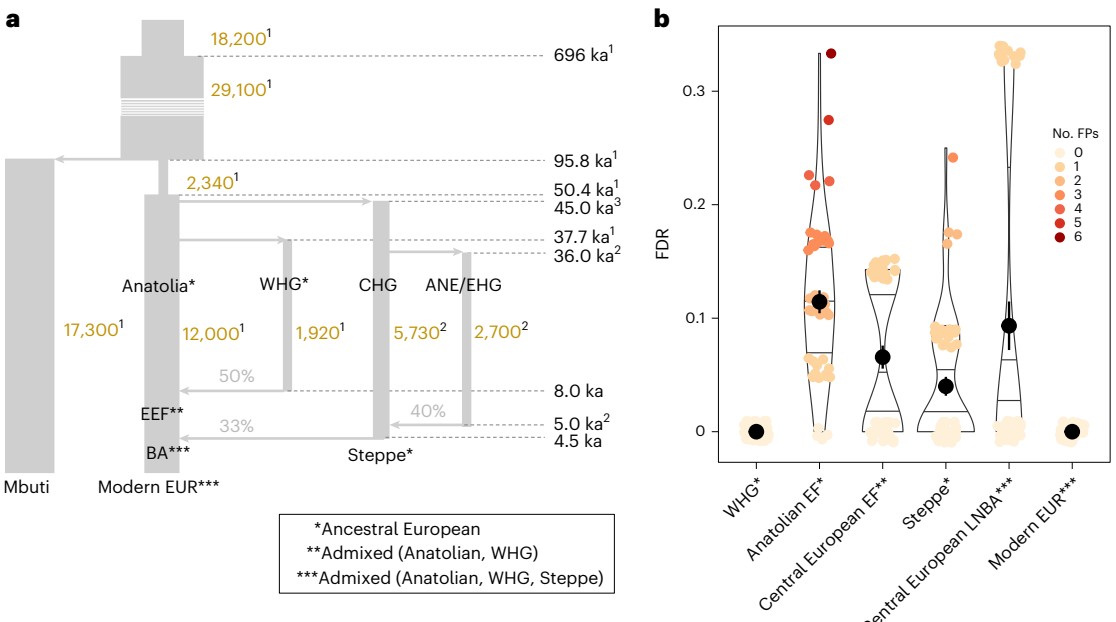

**Fig. 2 | Assessing the robustness of the hard sweep detection pipeline.**
**a**, Schematic of the West Eurasian population history model used to explore the statistical properties of our analytical pipeline and the impact of historical bottlenecks and admixture on the FDR. Each vertical segment denotes a major population branch (effective population sizes shown in gold text), with grey horizontal arrows denoting separation and admixture events (times shown on the right-hand side of the figure, assuming that admixture occurred 500 years after the onset of the migrations shown in Fig. 1; with percentages indicating the proportion of ancestry contributed by the incoming admixture branch). Model parameters are taken from one of three studies, as denoted by the associated superscript (1, ref. [41]; 2, ref. [42]; 3, ref. [43]), with CHG indicating Caucasus Hunter-Gatherers and ANE denoting Ancient North Eurasians. **b**, Estimated FDR measured at six different simulated populations sampled before (Anatolian EF, Steppe and WHG) and after major admixture events (EF, LNBA and Modern Europeans (EUR)). Results are shown for 30 simulated genomes, dots indicate mean values, horizontal lines represent quartile values (see Supplementary Fig. 19 for further information on sample size and sampling time), and the colour scale indicates the number of false positives (No. FPs). The maximum mean FDR observed amongst the simulated populations at this threshold, ~11%, was used as a conservative estimate for the study-wide FDR.

single-nucleotide polymorphism (SNP) scans; Methods) through a uniform bioinformatic pipeline, thereby mitigating potential processing artefacts, and grouped these samples into 18 distinct ancient populations that existed before and after documented Holocene admixture events based on their archaeological and genetic relationships (Fig. 1, Supplementary Figs. 1 and 2 and Supplementary Table 1). To ascertain the impact of Holocene admixture on modern European populations, we also analysed three modern European populations from the 1000 Genomes project[22] (specifically, Utah residents with northern and western European ancestry (CEU), Finnish in Finland (FIN) and Toscani in Italy (TSI)) and included two other non-European 1000 Genomes populations for further comparison (one East Asian: Han Chinese in Beijing (CHB); and one African: Yoruba in Ibadan (YRI)).

We used SweepFinder2 (SF2)[23,24] to scan the genomes of each ancient and modern population for regions exhibiting distorted allele frequency patterns characteristic of a fixed hard sweep (that is, an excess of low-frequency and high-frequency alleles[25]; see also Supplementary Text 2). By contrasting each window against site frequency spectrum (SFS) patterns from the overall genomic background, SF2 tests explicitly for the signature of fixed hard selective sweeps while controlling for demographic history, which often causes false positives in such tests[2,23]. A set of candidate sweep regions was determined by first assigning SF2 scores to a set of ~19,000 annotated human genes and subsequently aggregating neighbouring outlier genes into single sweep regions (Supplementary Fig. 3 and Methods). Importantly, testing of our sweep detection pipeline on simulated genomic datasets modelled on previously estimated Eurasian demographic history confirmed that it is robust to the effects of substantial demographic bottlenecks, including strong bottlenecks associated with the founding

Eurasian and subsequent WHG populations (successive 12-fold and sixfold population size reductions, respectively; Fig. 2a), and also to missing data, ascertainment bias and alignment error (Fig. 2b, Supplementary Figs. 4–7 and Supplementary Methods).

## Hard sweeps were common in ancient West Eurasian populations

In direct contrast to previous studies of modern human genomes[26], we were able to identify large numbers of hard sweeps in the ancient West Eurasian populations (Supplementary Figs. 8 and 9 and Supplementary Table 2), identifying 57 with high confidence (estimated study-wide false discovery rate (FDR) of <11% under a realistic simulated Eurasian demography; Fig. 2; Methods and Supplementary Methods). None of these sweep signals were evident in the YRI population, and ~90% of the sweeps showed significantly inflated levels of genetic divergence relative to African populations using $F_{st}$-based tests (outFLANK method[27]; Methods and Supplementary Fig. 10), consistent with the underlying selection pressures postdating the separation of the founding Eurasian group from African populations.

Although the estimated strength of selection was sufficient for these 57 putatively selected loci to have swept to high frequencies in West Eurasian populations by the Early Holocene to Bronze Age era (that is, ~12–5 ka; 41 sweeps having $s > 1\%$, with the largest value of $s$ nearing 10%; Supplementary Table 2; Methods and Supplementary Text 6), only two sweeps (1:35.4–36.5 and 6:29.5–32.8; Supplementary Figs. 8 and 9 and Supplementary Table 2) were still identified as SF2 outliers in selection scans of modern European populations. This dramatic reduction in hard sweep signals was not an artefact of differences in either data quality or quantity between modern and ancient

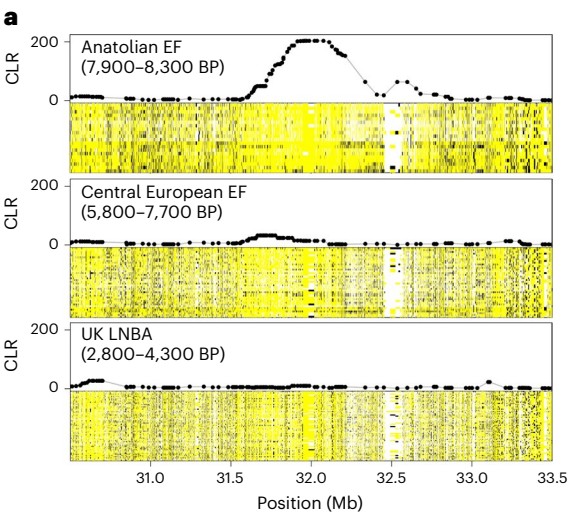

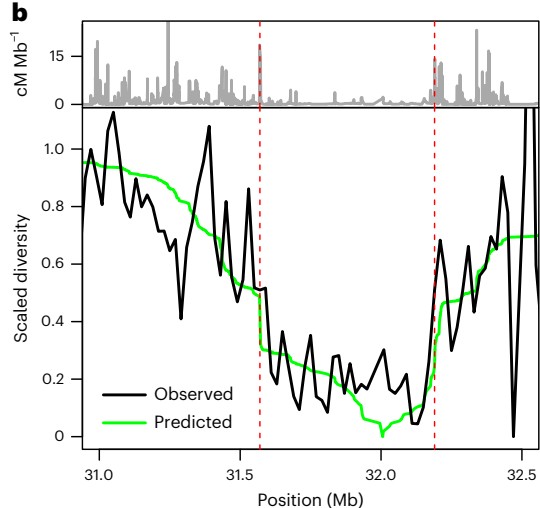

**Fig. 3 | Hard selective sweep in MHC-III region in Anatolian EF. a**, Haploimage of the MHC-III region and associated SweepFinder2 CLR score for the Anatolian EF, Central European EF and British Bronze Age (UK LNBA) populations. Pseudohaploid calls are shown for all samples in each population, with major alleles in yellow, minor alleles in black and missing data in white. Elevated SweepFinder2 CLR scores coincide with a region of depleted variation in the Anatolian EF population, which returns to background levels in the subsequent admixed populations. **b**, The estimated nucleotide diversity across the MHC-III region of the Anatolian EF (black line) and expected diversity under a hard sweep model (green line; Supplementary Text 3) relative to the underlying recombination rate in cM Mb$^{-1}$ (grey line on top). The two dashed red lines indicate local recombination hotspots that flank the sweep region. The close correspondence between the expected and observed genetic diversity across this region is unlikely to be a bioinformatic artefact and points to the authenticity of the signal.

populations (Supplementary Figs. 11–14 and Supplementary Methods). Nor could their absence be explained by the degradation of the sweep signals through random allele frequency changes (that is, genetic drift) and new mutations, as hard sweep signals in Eurasian human genomes are expected to remain visible to our analytical pipeline for around 70,000 years in the absence of other effects (Supplementary Fig. 15, Supplementary Methods and refs. [23,28]). Rather, the loss of the hard sweep signals is consistent with the reduction in sweep haplotype frequency below detection limits caused by the introduction of other orthologous haplotypes during the Holocene admixture phases. For the analytical pipeline used in this study, power drops below 50% if the sweep haplotype does not occur in at least 85% of the population at the time of sampling, and the sweep signal becomes effectively undetectable if less than half of the population has the sweep haplotype (Supplementary Fig. 15 and Supplementary Methods; see also ref. [29]).

## A historical hard sweep in the MHC-III region

Perhaps the most striking example of a previously unreported hard sweep in this study was provided by a ~1.5 Mb region overlapping the major histocompatibility complex class III (MHC-III) region on chromosome 6 that showed depleted genetic variation in the Anatolian EF population (Fig. 3). The MHC-III region includes numerous genes that encode antigen proteins involved in the immune response and has previously been identified as a recurrent target of balancing selection[30–32] and also as a potential source of selection artefacts owing to high levels of local genetic variation hindering read alignment across this region. However, nearly all (98.6%) SNPs in the MHC-III sweep region were found to lie in regions of high mappability, consistent with levels observed in the other 56 sweeps (minimum, 96.7%; Supplementary Table 3), indicating that poor read alignment was not a problem for our sweeps in general.

The authenticity of the MHC-III hard sweep is further supported by the distinctive trough of genetic diversity observed across the affected region in the Anatolian EF samples, which is located between two recombination hotspots (that is, genomic regions characterized by locally inflated recombination rates) that also demarcate the sharp transition to background diversity levels in either side of the sweep region (Fig. 3). This correlation between genetic diversity and local recombination rates is unlikely to arise from read misalignment artefacts but can be closely approximated by a hard sweep model acting on a centrally located beneficial variant that incorporates the local recombination rates (Fig. 3 and Supplementary Text 3), pointing to historical positive selection as the more likely cause.

In contrast to the Anatolian EF population, the two other major contributors to modern European genetic ancestry (that is, WHG and Steppe) had SF2 scores and patterns of local genomic variation that were indistinguishable from neutral background levels in the same ~1.5-Mb MHC-III region (Supplementary Fig. 16). Holocene-era populations show patterns that are consistent with a progressive dilution of the sweep signal across this period, with a weak SF2 signal still evident in EF populations (which draw ancestry from both Anatolian EF and WHG) that returns to neutral background levels in subsequent Bronze Age and modern European populations following the introduction of additional Steppe ancestry (Fig. 3 and Supplementary Fig. 16). Taken together, the evidence strongly implies that the MHC-III region was a target of strong positive selection in the ancestors of Anatolian EF and that the underlying hard sweep signal became masked in descendant populations by Holocene-era admixture involving genetically diverged populations.

## Admixture can obscure historical hard sweeps

The Anatolian EF-specific MHC-III hard sweep suggests that if Holocene admixture is responsible more generally for masking hard sweep signals in modern European genomes, then sweeps that were specific to only one of the admixing ancient Eurasian populations (for example, where selection is driven by a population-specific or local pressure) should be particularly prone to SF2 signal dilution from Holocene admixture events. Conversely, sweeps occurring closer in time to the Out-of-Africa migration are more likely to have signals that survive Holocene admixture events by virtue of being present at high frequencies in all admixing populations. To test whether sweep signal presence was dependent on the sweep antiquity, we inferred the approximate

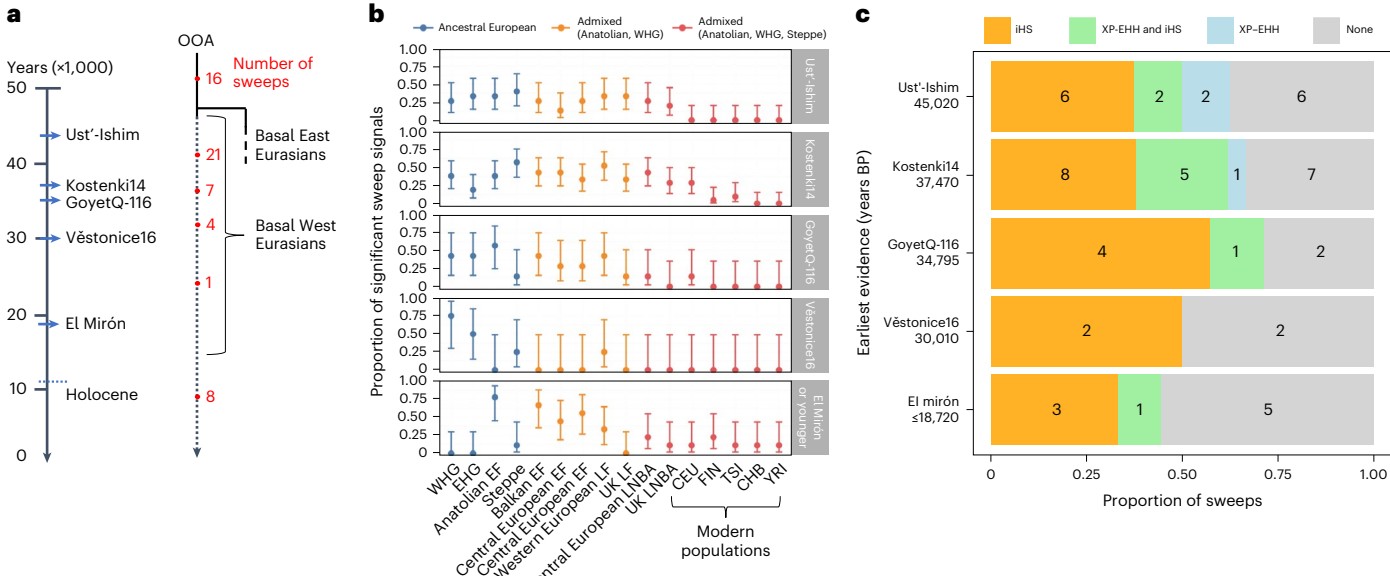

**Fig. 4 | Older sweeps are more robust to population admixture. a,** Schematic representation of the inferred temporal origins of the 57 sweeps. Each sweep was classified according to the first presence of the sweep haplotype amongst the five moderate-to-high-coverage Upper Palaeolithic specimens (italic labels, blue arrows indicate the approximate sample age), resulting in five distinct categories that are putative lower bounds of selection onset times: that is, Ust'-Ishim, *n* = 16; Kostenki14, *n* = 21; GoyetQ-116, *n* = 7; Věstonice16, *n* = 4; El Mirón, *n* = 9 (the final category also includes eight sweep haplotypes that were not observed in any Palaeolithic specimen). **b,c,** For each onset category, we quantified the proportion of sweeps observed for each tested population (**b**; dots indicate proportion of sweeps present at *q* < 0.05; error bars show 95% binomial confidence intervals) and classified sweeps according to results from two studies

reporting partial sweeps in modern Europeans[35,36] (**c**; integrated haplotype score, iHS; test statistics from ref. [35] being limited to outliers reported in at least two European populations to provide a stringent classification). Sweeps starting within the last 35,000 years (that is, not observed in GoyetQ-116 or older samples) tend to have patterns consistent with local selection, being highly frequent in some ancient populations but absent in others (Supplementary Text 4) and are less likely to be reported as partial sweeps nearing fixation (that is, lack an XP-EHH signal in ref. [35,36]; see key in **c**). Although the latter difference was not significant (one-sided Fisher's exact test *P* ~0.17), our results are consistent with sweeps arising after the diversification of the Eurasian founders being more susceptible to admixture distortion. CEU, Western European; Han Chinese in Beijing, CHB; FIN, Finnish in Finland; TSI, Toscani in Italy.

onset of the underlying selection pressure by manually scanning five moderate-to-high-coverage Upper Palaeolithic individuals for evidence of the sweep haplotype—namely, Ust'-Ishim ~45 ka[33], Kostenki14 ~37 ka[34], GoyetQ-116 ~35 ka[34], Věstonice16 ~30 ka[34] and El Mirón ~18 ka[34]—assigning each sweep to one of five age classes based on the oldest sample in which the sweep haplotype was observed (Fig. 4, Methods and Supplementary Text 4). Concordant sweep age classifications were obtained using an alternate quantitative method (based on a set of diagnostic marker alleles; Supplementary Fig. 17 and Supplementary Methods), suggesting that our classifications should provide robust inferences for aggregated sets of sweeps from the same age class, even though individual sweep dates are likely to be less reliable.

Sweep presence across the 12 ancient Eurasian populations was broadly consistent with the inferred antiquity of the selection pressure (Fig. 4b). Sweeps inferred to have started by 35 ka (that is, observed in GoyetQ-116 or an older sample) were detected at consistent levels across ancient populations both before and after the Holocene admixture events (no significant differences in sweep detection rates; logistic regression *P* > 0.17 for all three sweep age categories; Methods and Supplementary Text 4). By contrast, sweep detection differed significantly across the ancient populations for more recently selected loci (logistic regression *P* < 0.002 for both age categories) and exhibited patterns consistent with historical selection acting on populations ancestral to a subset of the tested populations (Fig. 4b; noting that the more pronounced signal loss evident in modern Europeans may reflect further dilution from admixture events following the Bronze Age; Discussion). Similarly, sweeps starting by 35 ka were more likely to appear as partial sweeps in two previous studies of positive selection in modern European populations[35,36] (66% versus 46% for sweeps

dated less than 35 ka; Fig. 4c) and were also more frequently observed as outliers for a statistic (XP-EHH) that is sensitive to loci nearing fixation (25% versus 8%; Fig. 4c). These results demonstrate that admixture can sufficiently distort the genetic signals resulting from ancient fixed sweeps, often leading to haplotype patterns in admixed populations that are misinterpreted as resulting from recent and potentially ongoing selection[28,35,37–39].

## Evolutionary scenarios underlying sweep signal dilution

To further assess the plausibility of Holocene admixture events masking historical hard sweep signals in modern human genomes, we used population genetic simulations (SLiM3)[40] to model selection (Supplementary Figs. 18–20; Methods and Supplementary Methods) within a plausible West Eurasian demographic model informed by ancient DNA studies[41–43] (Fig. 2a and Supplementary Fig. 19). Although our simulations might not capture the full complexity of West Eurasian demographic history, the model has sufficient detail to provide general insights into the effect of admixture on signatures of hard sweeps in these lineages. Beneficial mutations were introduced on the Main Eurasian branch before (55 ka) and after (44 ka and 36 ka) the diversification of the founding Eurasian population (Fig. 5 and Supplementary Figs. 19 and 20), with selection tests performed on the three ancient source populations (that is, Anatolian EF and WHG, both sampled at 8.5 ka, and Steppe sampled at 5 ka) and three admixed European populations descending from two separate Holocene admixture phases[44] that occurred at 8 ka (European EF, sampled at 7 ka) and 4.5 ka (Bronze Age Europeans, sampled at 4 ka, and Modern Europeans, sampled at 0 ka; Supplementary Text 1). All simulated datasets matched the sample

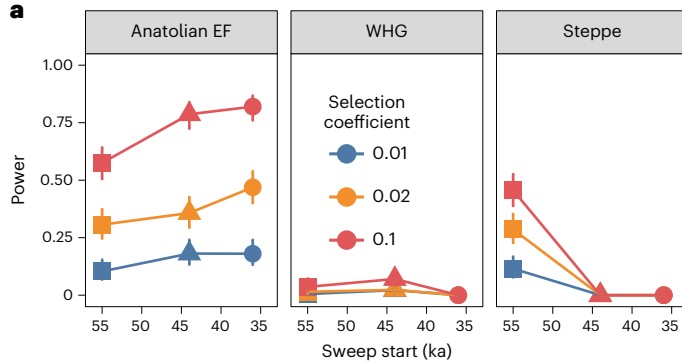

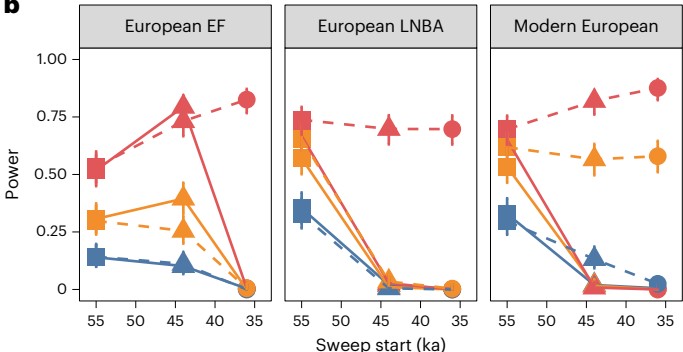

**Fig. 5 | Investigating the influence of admixture on sweep detection in modern populations. a,b,** Sweep detection power was estimated for selected loci simulated using a realistic Eurasian demographic model (Fig. 2a and Supplementary Fig. 19) with sample sizes based on empirical observations (that is, Anatolian EF, *n* = 28; WHG, *n* = 45, Steppe, *n* = 68; European EF, *n* = 78; European LNBA, *n* = 192; and Modern Europeans *n* = 200). Sweeps were timed to start before the diversification of the Eurasian founding population (55 ka) or following the separation of population branches that eventually gave rise to Steppe (44 ka) or WHG populations (36 ka). Mean power and 95% confidence intervals (measured at a FPR of 0.1%) are shown relative to the onset of selection in (**a**) the three European source populations (Anatolian EF, WHG and Steppe) as well as in (**b**) three admixed populations following the mixing of WHG and Anatolian EF at 8.5 ka (European EF, sampled at 7 ka) and the European EF and Steppe herder admixture at 4.5 ka (LNBA and Modern Europeans, sampled at 4 ka and 0 ka, respectively). Only beneficial mutations preceding the initial diversification of Eurasian lineages at 55 ka are evident in all populations when the selection pressure does not persist following the admixture events, with sweeps starting before 44 ka also detectable in European EF as they are shared by both source populations (**b**, solid lines). Notably, power increased appreciably for strongly selected loci (*s* ≥ 0.02) when selection was allowed to continue in the postadmixture phase (**b**, dashed lines) owing to these loci refixing following the admixture event.

sizes, SNP numbers, missing data and ascertainment bias observed in empirical data from the relevant populations (Methods).

We first investigated a model in which selection is active along all population branches that inherit the beneficial mutation until the 8 ka admixture event, at which point the selection pressure is relaxed (for example, owing to an environmental change or the emergence or introduction of a technological innovation proximal to the admixture event). Indeed, recent evidence suggests that spatiotemporal changes in selection pressures may have been common in recent human evolutionary history[45–49]. Our results clearly demonstrate that Holocene-era admixture can effectively mask historical sweep signals in the absence of any ongoing selection pressures: only the sweeps that precede the division of Eurasian lineages (that is, selection starting at 55 ka) could

be detected with reasonable power both before and after the admixture events. By contrast, sweeps starting after the diversification of Eurasian lineages (that is, 44 ka and 36 ka) had slightly increased power in unadmixed populations that experienced the selection pressure (consistent with having more recent fixation times and less signal loss from drift), whereas power was negligible in all admixed European populations (Fig. 5 and Supplementary Fig. 20). As expected, the detection rates were strongly positively correlated with selection strength, whereas sample size had a more moderate positive association.

Sweep detection power remained high in admixed populations whenever sweeps predated the split between the source population lineages, despite one of the source populations (WHG) having power close to zero under all modelled scenarios (Fig. 5 and Supplementary Fig. 20). The low power observed for WHG is likely to have been a consequence of the relatively small effective population size[41] exacerbating drift and causing rapid degradation of any fixed sweep signals (expected signal loss in $0.2 \times N_e$ generations[23] equates to ~11,000 years to loss in WHG assuming a generation time of 29 years[50], compared with ~70,000 years on the main Eurasian branch). This result indicates that hard sweeps are reasonably robust to admixture-induced signal loss when the sweep signal has been eroded in one of the source populations owing to drift, provided that the other source populations have retained the sweep signal.

After amending the model to allow the selection pressure to persist following the Holocene admixture events, we observed a dramatic increase in the detection rate of sweeps postdating the subdivision of Eurasian populations (that is, selection onset at 44 ka and 36 ka), with power being particularly high for strongly selected loci (*s* = 10%) across all admixed European populations (power between 65 and 85%; Fig. 5 and Supplementary Fig. 20). Notably, Modern Europeans achieved detection power exceeding that observed for any of the three ancestral source populations (Anatolian EF, Steppe and WHG) in nearly all cases. This result implies that many of the historical sweep signals should have been present in modern European populations had the underlying selection pressure persisted after the Holocene admixture phase, suggesting an attenuation of the underlying selection pressure(s), or the influence of other factors that were not included in the simulations, following the European Bronze Age (Discussion).

## The mutational basis of the Eurasian hard sweeps
Under canonical models of hard sweeps, beneficial mutations only emerge following the environmental shift marking the onset of the positive selection pressure, though recent theoretical and simulation-based work has shown that hard sweep signals are possible when selection acts instead on mutations segregating at the time of the environmental shift (that is, standing genetic variants; SGVs)[51–56]. Our results suggest that the underlying selection pressure(s) probably arose during the early stages of the Anatomically Modern Human occupation of Eurasia—for example, 16 (28%) and 44 (77%) of the 57 sweep haplotypes being observed in archaic samples dated at ~45 ka and ~35 ka, respectively (Fig. 4)—constraining the time in which de novo mutations could have arisen and suggesting that SGVs may have provided the mutational basis for many of our sweeps.

To examine this question in more detail, we used equations from ref. [57] to calculate the probability of a sweep arising from SGVs relative to de novo mutations, conditional on the fixation of the beneficial allele within a particular time interval under standard parameterizations (Supplementary Table 4 and Supplementary Text 5). Assuming that the environmental change triggering positive selection also coincided with a tenfold reduction in population size (that is, consistent with the Out-of-Africa bottleneck), we observed that fixation from SGVs was highly probable for positive selection strengths that matched the majority of our empirical estimates (*s* ~1%; Supplementary Table 2) when the temporal window in which fixation must occur was reasonably constrained (≤ 20 kyrs). By contrast, de novo variants dominated when

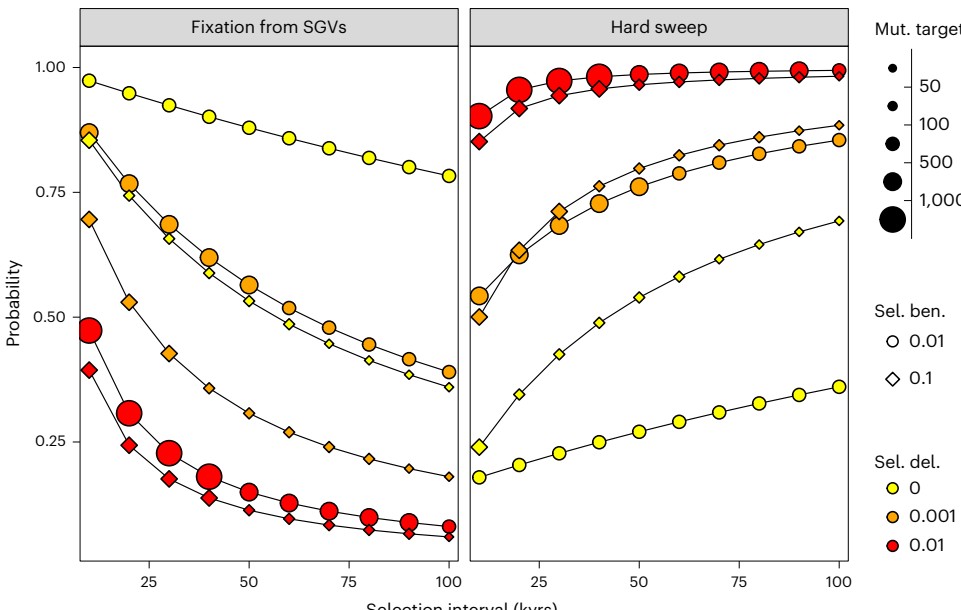

**Fig. 6 | Mutational origins of the Eurasian hard sweeps.** The probability that a sweep was caused by an SGV relative to a de novo mutation conditional on a sweep of either type occurring within a fixed time interval (left panel), following equations in ref. [57] (and assuming standard human generation times[50] and mutation rates[102]). SGVs are assumed to have been under some degree of purifying selection (denoted by different coloured symbols) before the environmental shift that leads to a strong bottleneck (that is, a tenfold reduction in population size; see Supplementary Fig. 21 for results from models with less severe bottlenecks) and initiates the beneficial selection phase (symbol shapes). Fixation from SGVs was highly likely (>75%) for constrained time intervals (≤20 kyrs) when the beneficial selection strength (Sel. ben.) was moderate (*s*

~0.01), both being plausible values for Eurasian sweeps, provided that purifying selection (Sel. del.) before the environmental shift was weak (*s* ≤ 0.001). Factoring in the probability that fixed SGVs all descend from a single copy at the time of environmental shift results in consistently high probabilities (≥50%) that selection resulted in a hard sweep pattern regardless of the mutational source, provided that SGVs had previously been deleterious (right panel). Notably, the fixation of variants that had previously been strongly deleterious required large mutational target (Mut. target) sizes (>1000 possible mutations) to ensure fixation within a 40,000-year interval when the beneficial selection strength was ~0.01, which may be implausibly large for many traits.

purifying selection on SGVs was strong (*s* ≥ 1%) in the period preceding the environmental shift (Fig. 6 and Supplementary Fig. 21). Extending our calculations to include the probability that selection from SGVs produces a hard sweep signal[57] (that is, only one of the beneficial haplotypes present at the time of the environmental shift ultimately fixes) revealed that hard sweeps are highly probable (>50%) regardless of the mutational input source, provided that SGVs have been deleterious in the period preceding the environmental change (Supplementary Figs. 6 and 21 and Supplementary Text 5). Although further distinguishing between the two modes of selection was hampered by the absence of prior information on several key parameters (for example, the strength of purifying selection on SGVs, the type and strength of epistasis and the distance to the new fitness optimum), our results imply that mutations underlying the hard sweeps were probably initially deleterious and reinforce previous findings showing that selection from both de novo mutations and SGVs have occurred in Eurasian evolutionary history[53].

## Discussion

Our analyses of >1,000 ancient West Eurasian genomes has uncovered strong evidence for 57 hard sweeps in Early Holocene to Middle Holocene populations that have been almost entirely erased from descendent populations in modern Eurasia. Most of these selected loci had probably swept to high frequencies well before the Holocene era; this is supported by the moderate-to-strong selection strengths that were inferred for the sweeps. These selection coefficients are comparable with the strongest currently known for human populations (that is, the *LCT* locus having *s* between ~2% and 6% in the recent history of European populations[58,59]) and suggest that such strong positive selection events have been much more common in recent human history than previously recognized.

Our empirical and simulation results implicate Holocene-era admixture as the primary factor attenuating these historical sweep signals, which has led to them being missed in previous studies or detected instead as empirical genome-wide outliers of haplotype-based selection statistics[35,36]. Such haplotype-based outliers are typically interpreted as ongoing partial sweeps resulting from recent selection[28,35,37–39], as the underlying haplotype patterns decay quickly over time and become largely undetectable for selection starting more than 30,000 years ago in humans[28,39]. However, our analyses suggest that most (85%) of the 41 sweeps that overlap with a haplotype-based outlier were already under selection by 30 ka, implying that these signals are more likely to result from admixture-driven dilution of hard sweeps that mostly began before 30,000 years ago.

An intriguing implication arising from the simulations is that the selection pressure(s) underlying the sweeps may have eased during the Holocene period in some cases. This period marked the introduction of new technologies and diets, a stable warm climate and living conditions that introduced new selection pressures (for example, selection on the *LCT* gene to reduce the costs associated with milk consumption in adulthood[59]) and may have also reduced the intensity of historical selection pressures underlying the 57 hard sweeps. Alternatively, these patterns could have resulted from the introduction of new ancestry sources into Europe following the Bronze Age period[60–62], which would have further diluted sweep signals and left insufficient time for their reappearance even if the selection pressure was still present. Indeed, this may explain why modern Tuscans and Finns had the fewest sweeps of any Eurasian population examined in this study, with both populations descending from post-Bronze Age mixing events that introduced distinct ancestry components (from Northern African[60,61] and Siberian[62] groups, respectively) that were not apparent in the other surveyed Eurasian populations (Fig. 4b).

Although our analyses point to well-known admixture events during the Holocene as the prime driver of the diluted sweep signals observed in modern European genomes, it is possible that the three populations directly ancestral to present-day Europeans (that is, Mesolithic hunter-gatherers, Anatolian Neolithic farmers and pastoralists from the Pontic–Caspian steppe) were also admixed to some degree[63]. However, the much stronger genetic differentiation observed between the three ancestral populations relative to the later Holocene European groups[21] suggests that potential admixture events involving the three ancestral lineages were probably less influential or frequent than subsequent admixture phases in the Holocene[21]. This implies less perturbation of any underlying sweep signals, although we note that the occurrence of such mixing events would mean that historical hard sweeps were even more frequent than identified in our study.

In addition to masking historical sweeps in human populations, the obscuring effect of admixture might explain why species-wide selective sweep signals are rare in many species[9,64] while being abundant in others[65–67]. Crucially, recently admixed populations often lack detectable signs of structure, in which case admixture will not be correctly accounted for in any subsequent selection tests. For example, modern European populations have been considered to be sufficiently genetically homogeneous for the purpose of selection scans[8], despite ancient DNA studies revealing that Europeans have multiple diverged ancestry components[21]. Indeed, modern genomic data is often insufficient to establish past admixture events, with widely used principal component analysis (PCA) and ancestry decomposition methods being unable to detect historical admixture signals when suitable proxy populations for the admixing groups are lacking. Similarly, although admixture creates temporal variation in genomic coalescence rates, these patterns can be equally well explained by historical population size changes in a single continuous population[68]. Accordingly, species with little apparent population structure may be more susceptible to the confounding effects of admixture on selection scans than those in which structure is evident and can be directly accounted for, and this may partly explain why species with distinctive population structuring often show strong local selection signals (for example, Swedish *Arabidopsis thaliana*[3], African *Drosophila melanogaster*[69] and *Microbotryum lychnidis-dioicae*[70]), whereas genetically homogeneous taxa or populations tend to lack fixed hard sweeps but harbour abundant partial or soft sweep signals (for example, European humans[8,35], North American *D. melanogaster*[6,71] and *M. silenes-dioicae*[70]).

In accordance with previous work[3,4,18], our results emphasize the importance of incorporating historical population structure and admixture events into the null models of selection tests. If this information is not available—for example, because DNA from ancestral source populations is lacking, the default scenario for most species—then the interpretation of historical selection signatures could be highly misleading and heavily weighted against the detection of historical hard sweep events. Although these factors imply that the extent of past hard sweep events has probably been underestimated in natural populations in general, we caution that they do not directly challenge results from previous studies proposing a substantial role for soft sweeps in the adaptive history of humans[8] and other species[71]. Rather, we reiterate the conclusions of other recent work[6,72] that advocate for improved modelling of complex but widespread evolutionary and demographic processes to achieve an unbiased understanding of the mode and tempo of adaptation in natural populations.

## Methods

### Population designation

All ancient individuals used in this study were assigned to historical populations based on published analyses of their genetic relationships in combination with details regarding their archaeological context, with the temporal and spatial variability between individuals being minimized where possible (Supplementary Figs. 1 and 2 and

Supplementary Table 1). Our sample grouping resulted in 18 ancient populations occurring before and after the major Holocene admixture events that created the genetic landscape of modern Europe[19,21] (Supplementary Text 1). Additional testing using alternate sets of samples for two ancient populations showed that sweep signal detection was reasonably robust to changes in population sample configuration (Supplementary Methods and Supplementary Figs. 13 and 14).

### Data collection and processing

To produce a robust dataset and avoid potential bioinformatic batch effects, the raw sequence read data for 1,162 ancient genomic datasets (Supplementary Table 1) was retrieved from publicly available repositories (Short Read Archive: SRP029640, SRP057056; European Nucleotide Archive: ERP003900, PRJEB6622, PRJEB6272, PRJNA230689, PRJEB7618, PRJEB609, PRJEB9021, PRJEB8987, PRJEB9783, PRJEB11364, PRJEB11450, PRJEB1418, PRJEB13123, PRJEB11848, PRJEB14455, PRJEB22629, PRJEB12155, PRJEB22652, PRJEB23635, PRJEB24794, PRJEB29603) and processed through the following standardized pipeline. To minimize the risk of modern contamination, the forward and reverse reads of the paired-end reads were merged (collapsed) using fastp[73], and only merged reads were retained (modern data are more likely to comprise large DNA fragments that do not collapse). All collapsed reads were filtered for potential residual adaptor sequences and chimaeras using Poly-X with fastp[73]. The retained filtered set of sequence reads were aligned to the human reference genome (h37d) using the Burrows-Wheeler Aligner v.0.7.15 (ref. [74]). All mapped reads were sorted using SAMtools v.1.3 (ref. [75]) and then realigned around insertions and deletions, and potential PCR duplicate reads were marked and removed using the Genome Analysis ToolKit v.3.5 (ref. [76]).

Before variant calling, all remaining aligned reads were screened and base-calls were recalibrated for ancient DNA (aDNA) postmortem damage using mapDamage2 (ref. [77]). To further limit the impact of postmortem damage on variant calling[78], bamUtil[79] was used to trim three base pairs from each of the 5′ and 3′ ends of each mapped read. From the resulting set of reads, pseudohaploid variants were called at the set 1240k capture SNPs[80] found on the 22 autosomes, using a combination of SAMtools mpileup[81] and sequenceTools (https://github.com/stschiff/sequenceTools). Pseudohaploidization of read data is a standard strategy in aDNA analyses, whereby a single read is randomly sampled at each prespecified SNP position[80] to mitigate potential biases introduced by differences in coverage or postmortem damage between samples[19]. The 1240k capture was developed to minimize ascertainment in non-African populations and was used to generate data for most samples used in the study, whereby concentrating on the 1240k variants ensured a common and robust set of variants for the subsequent analyses. The pseudohaploid variant calls were converted from EIGENSOFT format[82,83] to binary PLINK format using EIGENSOFT. PLINK v.1.9 (ref. [84,85]) was used to assign samples to the predefined populations (Supplementary Table 1) and convert the variants to reference-polarized VCF files, with correct polarization being checked using BCFtools[81]. Finally, a custom Python script was used to generate the SFS input files for SF2 analysis (https://gist.github.com/yassineS/fe2712ad52d76460b927e3f391ea51f6).

### Sweep scans

The SweepFinder2 composite likelihood ratio (CLR) statistic[23,24] was computed across successive 1-kb intervals across all autosomes for each ancient and modern human population. The CLR statistic evaluates the evidence for hard selective sweeps in dynamically sized windows by calculating the expected SFS under a hard selective sweep conditional on the neutral SFS, assuming a certain selection coefficient and local recombination rate. The neutral SFS is based on the 'background' SFS calculated from the whole genome assuming that the influence of positive selection on the genome-wide SFS is negligible.

SF2 controls for genome-wide effects such as ascertainment bias and demographic history by allowing these processes to affect the expected SFS under neutrality (that is, the background SFS)[86]. Further, unlike many other selection methods, the assumptions on the input data for SF2 are suitable for the low coverage and ascertained nature of ancient DNA datasets. As it is only based on the spatial (genomic) pattern of allele frequencies but not on haplotype homozygosity or population differentiation, it is possible to detect selection without reference to a second population, calling genotypes or phasing haplotypes. Leveraging an empirical null model (that is, the background SFS) and a model-based alternative hypothesis, SF2 is both more powerful and more robust than alternate test statistics also based on deviations of the SFS from expectations under the standard neutral model (for example, Tajima's *D*, Fay and Wu's *H*)[23,24].

Note that SF2 also has an option to detect sweeps based on local genomic reductions in diversity. However, we did not calculate this diversity-based metric as the accurate and unbiased estimation of diversity requires fully sequenced genomes, whereas our dataset consisted of an ascertained set of SNPs.

### Outlier gene detection

Human gene annotations were obtained from the ENSEMBL database[87] (genome reference version GRCh37), which was accessed using the R biomaRt package[88,89] (v.2.36.1). Of the 24,554 annotated 'genes' on the biomaRt database, we removed any that were not annotated in the NCBI database (ftp://ftp.ncbi.nih.gov/gene/DATA/GENE_INFO/Mammalia/Homo_sapiens.gene_info.gz) and also excluded those that lacked specific protein-based and RNA-based annotations (in the biomaRt transcript_biotype field). This resulted in a list of 19,603 genes, from which we removed 26 that did not contain any polymorphic sites in our datasets (all being situated in the most terminal areas of chromosomes), leaving 19,577 genes that were used in the subsequent analyses (Supplementary Table 2).

To obtain *P* values for each gene, we transformed the raw Sweep-Finder2 CLR scores to *z* scores using the following series of steps (Supplementary Fig. 3). For each population, all SweepFinder2 CLR scores were logarithmically transformed and assigned to each of the 19,577 genes by binning the transformed scores within the genomic boundaries of each gene. The gene boundaries were extended by 50 kb on either side to also capture *cis*-regulatory regions. As this typically resulted in several scores per gene, we took the maximum score to represent the evidence for a sweep involving that gene. Each gene score was corrected for gene length using a nonparametric standardization algorithm[90,91], resulting in the gene scores having an approximately standard normal distribution (Supplementary Fig. 4). Finally, *P* values were calculated for all genes, and a *q* value correction[88] was applied for each population. The *q* value is a Bayesian posterior estimate of the *P* value that accounts for the expected inflation of false positives due to multiple testing[88], whereby a *q* value of 0.01 implies an FDR of 1% per population in this study.

### Candidate sweep classification

Sweeps were identified by determining a set of outlier genes across all populations, which were classified into sweep regions according to (1) the distance between the midpoint of neighbouring pairs of outlier genes (the intergene distance) and (2) overlapping sweep regions between populations. Step 1 was performed independently for each population, whereby all outlier genes with midpoints that were less than a specific distance apart from the midpoint of a neighbouring outlier gene were collapsed into a single category. After generating the collapsed categories for each population, step 2 was applied to ensure that the sweep categories sharing at least one gene across different populations were considered as a single historical sweep.

We ran our sweep quantification pipeline at three *q* value thresholds (*q* < 0.01, 0.05 or 0.10; which imply FDR values of 1%, 5% or 10%

per population, respectively) and three different intergene distances (midpoint distances less than 250 kb, 500 kb or 1 Mb). As expected, changing the *q* value had a large impact on the number of sweeps (ranging from ~50 for *q* < 0.01 to ~500 for *q* < 0.1), whereas changing the intergene midpoint distance had comparatively little impact overall, particularly at more stringent *q* value cutoffs (Supplementary Fig. 8). Based on these results, we decided to use the most stringent *q* value cutoff and the most liberal intergene distance to define a conservative set of candidate sweeps for all further analysis. However, because this stringent cutoff might lead to the removal of potentially causal genes in a sweep (which could have values slightly lower than 0.01), we first defined our sweeps based on the more permissive *q* < 0.1 threshold and then removed all sweeps that did not have at least one gene with *q* < 0.01. To further improve sweep determination, we removed populations with small sample sizes (Impact of sample size and missing data on sweep detection) from the sweep determination process, as previous analyses of modern genomes suggest that SF2 has little power to detect sweeps when the number of haploid genome copies being analysed is 10 or less[29]. Finally, to ensure that the sweeps being defined were all relevant to western Eurasian history, the modern populations from East Asia (CHB) and Africa (YRI) were also excluded from the sweep classification process. This strategy resulted in a total of 57 candidate sweeps that were used in subsequent analyses.

### Impact of sample size and missing data on sweep detection

Previous results suggest that SF2 maintains high power to detect sweeps when the number of haplotypes being analysed is 10 or more for modern genomes[29]. To examine the impact of sample size on SF2 estimation on our ancient populations, we derived a measure of sample size that incorporates both pseudohaploidy and different levels of data missingness in our ancient samples, called the 'effective' sample size, $n_{eff}$. We calculated $n_{eff}$ for each population as $kn(1 − M)$, where $k$ is the ploidy of each sample, $n$ is the number of samples and $M$ is the average proportion of missing sites at informative SNPs in that population. Consistent with results from modern data[29], we found that the number of detected sweeps tended to be systematically lower for populations with $n_{eff}$ values <10, regardless of the *q* value threshold used (Supplementary Figs. 8 and 12). When inspecting the distribution of *P* values for gene scores in each population, we observed large deviations from the idealized distribution as $n_{eff}$ values decreased below 10 (Supplementary Figs. 4 and 5). These results indicate that populations with $n_{eff}$ < 10 lack sufficient power to detect sweep signals, whereby only the 12 ancient populations with $n_{eff} \geq 10$ were used to determine the candidate sweeps.

### Selection strength inference

We inferred the selection strength, *s*, for each of our 57 candidate sweeps as $s = r \ln(2N_e)/\alpha$ (from ref. [92]), where $N_e$ is the effective population size, *r* is the recombination rate and $\alpha$ is a composite selection parameter that is estimated for each sweep region by SF2 (ref. [23]) (Supplementary Table 2). For each candidate sweep, we took $\alpha$ at the position of the largest SF2 CLR value from all ancient Eurasian populations. Recombination rates were estimated for each sweep using information from ref. [93], and $N_e$ was taken as 10,000 (noting that the estimation of *s* is robust to changes in $N_e$ owing to the logarithmic transformation). Notably, estimating *s* based on SF2 $\alpha$ estimates tended to result in slightly lower values than expected (Supplementary Methods and Supplementary Fig. 18), suggesting that we have probably systematically underestimated the strength of selection for the 57 observed sweeps.

### Estimating the earliest evidence for selection

To estimate when the selection pressure(s) underlying each of the 57 sweeps may have first arisen, we manually inspected five Upper Palaeolithic Eurasian human samples with moderate-to-high-coverage genomes (Ust'-Ishim[94], Kostenki14 (ref. [95]), GoyetQ-116 (ref. [34]), Věstonice[34] and El Mirón[34]) for evidence of the sweep haplotype.

The oldest sample with evidence for the sweep was taken as the origin of the selection pressure and therefore provided a coarse lower bound on the onset of selection. The sweep was called as present only if the full set of alleles observed in the sweep region was observed in the sample, noting that genotype calls were only possible for Ust'-Ishim (coverage >40×), with pseudohaploid calls being used for the other four samples. As some sweeps contained complex signals with multiple peaks—which may represent the combination of two or more neighbouring selection events into a single sweep region (see Supplementary Table 2 for a list of sweeps and Supplementary Data 1–57 (ref. [96]) for the distribution of SF2 CLR scores for each sweep region across all tested populations)—we concentrated on alleles underlying the strongest signal in each sweep. The plots used to discriminate the sweep haplotypes for all 57 candidate sweeps are provided in Supplementary Data 58–114 (ref. [97]). To test the robustness of these qualitative assessments, we compared our qualitative classifications with those obtained from a quantitative method. Details of this method and the comparison with qualitative results are provided in the Supplementary Methods.

### Sweep detection rate relative to inferred onset of selection

We predicted that hard sweep signals would be more prone to admixture-mediated loss if the onset of the selection pressure post-dated the initial diversification of the Eurasian founding populations, as these sweeps are more likely to be 'local' to a subset of the source populations contributing to modern European ancestry. Accordingly, we reasoned that sweeps of deep antiquity should exhibit consistent detection frequencies across all ancient populations used in this study, whereas more recent sweeps should show interpopulation variation that reflects their restriction to a subset of historical source populations. To examine this hypothesis, we quantified the proportion of sweeps present in each ancient population conditional on the inferred age of the sweep (Methods: Estimating the earliest evidence for selection) and then used the glm function in the R statistical language to test whether proportions differed significantly across ancient populations grouped into five broad categories (HG, EF, Steppe, LF and LNBA; Fig. 4). Specifically, for each sweep age class, we fitted a logistic regression model where the presence of the sweep was a binary dependent variable and population group was the sole independent variable and compared the fit of this model with the null model (dependent population variable removed; that is, all groups had the same proportion of sweeps) to compute $P$ values. Note that modern European populations were not included in these tests, as our analyses indicated that these populations experienced further signal dilution following the Bronze Age (Supplementary Text 1). Our results were unchanged after repeating all logistic regression analyses with the Southern Caucasus LNBA population removed from the LNBA group (as this population showed different ancestry composition to other European LNBA populations; Supplementary Methods).

### $F_{st}$-based selection tests

To further investigate the validity of the 57 candidate sweeps, we tested whether our sweeps were enriched with highly divergent SNPs amongst the 12 ancient Eurasian populations with sufficient power to reliably detect sweeps (Methods: Impact of sample size and missing data on sweep detection) with a modern African population (YRI). We calculated $F_{st}$ for each of the ~1.1M ascertained SNPs using the standard Weir–Cockerham estimator[98] and used OutFLANK[27] to estimate the probability that each SNP was more divergent than expected under neutrality (based on estimating fitting a $\chi^2$ distribution to $F_{st}$ values from putatively neutral SNPs). Importantly, OutFLANK is robust to non-equilibrium demographic models[27], including rapid range expansions that are thought to have resulted in highly divergent alleles observed in modern human populations (that is, through allele surfing[99]). Applying the $q$ value correction to the $P$ values to control for multiple testing resulted in 29 of the 57 candidate sweeps having one or more SNPs

with $q < 0.05$ and 55 candidate sweeps having at least one SNP with $q < 0.20$ (Supplementary Fig. 10). We then tested whether the SNPs found in each of the 57 candidate sweeps had significantly higher $F_{st}$ values than background genome levels; 49 of the 57 sweeps had significantly elevated $F_{st}$ values relative to the remaining background genome ($P < 0.05$; one-sided Wilcoxon rank-sum test; Supplementary Fig. 10), confirming that our sweeps were most likely to have been selected following the divergence of ancestral Eurasians from African populations and were unlikely to be artefacts caused by nonequilibrium demographic processes such as allele surfing.

### Forward simulations

To assess the impact of Holocene admixture on historical hard sweep signals in modern human genomes, we used forward simulations (SLiM3)[40] to model moderate-to-strong selection (selection coefficient $s$ between 1% and 10%) within a plausible West Eurasian demographic model (Fig. 2a and Supplementary Fig. 19). Model parameters were largely taken from a recent study of Eurasian demographic history[41], with further parameters for the Steppe population coming from two additional studies incorporating ancient Steppe samples[42,43]. Two Holocene-era admixture events were included, namely, a 50% contribution of WHG ancestry into the Main Eurasian branch at 8 ka and a further 33% contribution of Steppe (Yamnaya[42]) ancestry onto the same branch at 4.5 ka. These admixture events generated the typical modern European individual that derives approximately one-third of their ancestry from each of the Anatolian EF (modelled as descending from the Main Eurasian branch), WHG and Steppe populations[19].

Beneficial mutations were introduced on the Main Eurasian branch at three different times: 55 ka, 44 ka and 36 ka (Supplementary Fig. 19). We investigated a model where the strength of selection ($s$ either 1%, 2% or 10%) was constant until the present and another model where selection ceased following the 8 ka admixture event ($s = 0$ after 8 ka). Genomic regions of 5 Mb were simulated assuming mutation rates and recombination rates of $1 \times 10^{-8}$ per generation per base pair, with the beneficial variant being introduced as a single mutation in the middle of the simulated region.

Diploid samples were generated for one modern population, sampled at the present, and five ancient populations: three populations that are ancestral to modern Europeans (Anatolian EF and WHG, sampled at 8.5 ka, and Steppe, sampled at 5 ka) and two admixed populations (Central Europe EF and Central Europe LNBA, sampled at 7 ka and 4 ka, respectively). Sample sizes matched those in empirical data, with 100 diploid samples taken for the modern Europeans. For each ancient population, we simulated missing data by randomly removing loci based on the site-specific missing data distribution of the relevant empirical samples, thereby replicating the population-specific data missingness patterns. We also reproduced the pseudohaploidy of the ancient samples by randomly selecting one allele at each site. In addition, because our analyses were limited to a set of ~1.1 million variants from the 1240k capture probes (Methods: Data collection and processing), which were selected based on heterozygosity in a set of sequenced reference individuals from both African and non-African populations[100], we also sought to reproduce the ascertainment bias associated with these variants. To do this, we sampled two modern African samples and only retained sites that were heterozygous in either of the two African samples or at least one of two randomly chosen modern European samples, with all other sites being discarded.

Finally, in all simulations, the number of variant sites was downsampled to 2,000 positions in accordance with the average number of variants from 1240k capture probes found in a 5-Mb region. After removal of all nonpolymorphic sites, the SF2 statistic was computed at 5,000 positions evenly distributed over the 5-Mb region. To compute the neutral background SFS necessary for the SF2 analyses, we simulated 1,000 neutral replicates of the demographic model and extracted the SFS for each population after replicating all steps in our

simulation pipeline. Two hundred replicates were generated for each selection scenario and used for power analyses and to estimate the false positive rate (FPR). The final set of selection simulations were conditioned on sweeps that had escaped the initial stochastic phase (where rare beneficial alleles are lost through drift) by omitting all simulations where the beneficial mutation never reached a frequency of at least 10%. More details on the statistical modelling procedure and full specification of the power analyses and FPR estimation are provided in the Supplementary Methods.

### Reporting summary

Further information on research design is available in the Nature Research Reporting Summary linked to this article.

## Data availability

Allele frequency data and SweepFinder2 results for individual ancient populations[101] can be accessed under https://doi.org/10.25909/6324956ee6ba6. Supplemental Data 1–57 (ref. [96]) and 58–114 (ref. [97]) are accessible under https://doi.org/10.25909/631595cfd2733 and https://doi.org/10.25909/631595df4f1b2, respectively.

## Code availability

The custom script used to compute SFS from VCF files is available at https://gist.github.com/yassineS/fe2712ad52d-76460b927e3f391ea51f6. All other custom R scripts used in the analysis of empirical and simulated data are available from the corresponding authors upon request.

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

## Acknowledgements

We thank F. Racimo, I. Mathieson, W. Haak, J. Pritchard, N. Bean, B. Llamas, L. Weyrich, J. Breen, J. Soubrier, C. Franks, C. Miller, M. Brooks and R. Ward (deceased). We gratefully acknowledge the museums, archaeological collections, collectors and curators who were responsible for the original skeletal material and made samples available for genomic analysis. Funding was received from Australian Research Council Grants to A.C. (FL140100260), C.D.H. (DE180100883), and R.T. (DE190101069); the Australian National Database Service; Nectar Cloud; and the University of Adelaide Environment Institute, Bioplatforms Australia.

## Author contributions

A.C., Y.S., R.T. and C.D.H. conceived the study; Y.S., R.T. and M.W. assembled the dataset; Y.S., R.T., A.J., C.D.H., G.G., M.C., A.R. and O.J. performed analyses; A.C. and C.D.H. supervised analyses; Y.S., R.T., A.J., C.D.H., S.G., C.T., A.R., O.J., J.S., J.T. and A.C. interpreted the results; and A.C., Y.S., R.T., A.J. and C.D.H. wrote the paper with input from all coauthors.

## Competing interests

The authors declare no competing interests.

## Additional information

**Correspondence and requests for materials** should be addressed to Yassine Souilmi, Raymond Tobler, Angad Johar, Alan Cooper or Christian D. Huber.

# Reporting Summary

## Statistics

For all statistical analyses, confirm that the following items are present in the figure legend, table legend, main text, or Methods section.

| n/a | Confirmed | |
|---|---|---|
| ☐ | ☒ | The exact sample size ($n$) for each experimental group/condition, given as a discrete number and unit of measurement |
| ☐ | ☒ | A statement on whether measurements were taken from distinct samples or whether the same sample was measured repeatedly |
| ☐ | ☒ | The statistical test(s) used AND whether they are one- or two-sided<br>*Only common tests should be described solely by name; describe more complex techniques in the Methods section.* |
| ☐ | ☒ | A description of all covariates tested |
| ☐ | ☒ | A description of any assumptions or corrections, such as tests of normality and adjustment for multiple comparisons |
| ☐ | ☒ | A full description of the statistical parameters including central tendency (e.g. means) or other basic estimates (e.g. regression coefficient) AND variation (e.g. standard deviation) or associated estimates of uncertainty (e.g. confidence intervals) |
| ☐ | ☒ | For null hypothesis testing, the test statistic (e.g. $F$, $t$, $r$) with confidence intervals, effect sizes, degrees of freedom and $P$ value noted<br>*Give P values as exact values whenever suitable.* |
| ☒ | ☐ | For Bayesian analysis, information on the choice of priors and Markov chain Monte Carlo settings |
| ☒ | ☐ | For hierarchical and complex designs, identification of the appropriate level for tests and full reporting of outcomes |
| ☐ | ☒ | Estimates of effect sizes (e.g. Cohen's $d$, Pearson's $r$), indicating how they were calculated |

*Our web collection on statistics for biologists contains articles on many of the points above.*

## Software and code

Policy information about availability of computer code

| Data collection | Data were manually curated from publicly available datasets. |
|---|---|
| Data analysis | Custom R and SLiM scripts used to perform analyses in this study are available from corresponding authors upon request. |

For manuscripts utilizing custom algorithms or software that are central to the research but not yet described in published literature, software must be made available to editors and reviewers. We strongly encourage code deposition in a community repository (e.g. GitHub). See the Nature Portfolio guidelines for submitting code & software for further information.

## Data

Policy information about availability of data

All manuscripts must include a data availability statement. This statement should provide the following information, where applicable:
- Accession codes, unique identifiers, or web links for publicly available datasets
- A description of any restrictions on data availability
- For clinical datasets or third party data, please ensure that the statement adheres to our policy

Access codes for all data used in the project are provided in the Data Availability statement. A complete description of how the raw sequence data were processed and prepared for analysis is provided in the Methods.

# Field-specific reporting

Please select the one below that is the best fit for your research. If you are not sure, read the appropriate sections before making your selection.

☐ Life sciences  ☐ Behavioural & social sciences  ☒ Ecological, evolutionary & environmental sciences

For a reference copy of the document with all sections, see nature.com/documents/nr-reporting-summary-flat.pdf

# Ecological, evolutionary & environmental sciences study design

All studies must disclose on these points even when the disclosure is negative.

| | |
|---|---|
| Study description | Population genomic analyses of hard sweep signals in ancient and modern human genomes with a focus on historical Eurasian populations. Putative hard sweeps were inferred in each tested population using SweepFinder2 method, with 57 candidate sweeps being determined in total. Additional analyses explore the evidence for hard sweep signal loss following known Holocene admixture events. |
| Research sample | Samples were chosen from all available Eurasian aDNA samples available at the commencement of the study and placed into 'populations' based on archaeological and population genetic criteria. The full rationale for the population groupings is described in Methods and SI. |
| Sampling strategy | Rationale for population groupings is described in Methods and SI. |
| Data collection | NA: study uses previously published data. |
| Timing and spatial scale | Rationale for population groupings is described in Methods and SI. |
| Data exclusions | Populations with small effective sample sizes were excluded from some analyses. The rationale is explained in the Methods and SI. |
| Reproducibility | All sequence processing and bioinformatic and statistical methods used in the study are reported in the Methods and SI. |
| Randomization | Rationale for population groupings is described in Methods and SI. |
| Blinding | NA: study uses previously published data. |

Did the study involve field work? ☐ Yes ☒ No

# Reporting for specific materials, systems and methods

We require information from authors about some types of materials, experimental systems and methods used in many studies. Here, indicate whether each material, system or method listed is relevant to your study. If you are not sure if a list item applies to your research, read the appropriate section before selecting a response.

## Materials & experimental systems

| n/a | Involved in the study |
|---|---|
| ☒ | Antibodies |
| ☒ | Eukaryotic cell lines |
| ☐ | ☒ Palaeontology and archaeology |
| ☒ | Animals and other organisms |
| ☒ | Human research participants |
| ☒ | Clinical data |
| ☒ | Dual use research of concern |

## Methods

| n/a | Involved in the study |
|---|---|
| ☒ | ChIP-seq |
| ☒ | Flow cytometry |
| ☒ | MRI-based neuroimaging |

# Palaeontology and Archaeology

| | |
|---|---|
| Specimen provenance | All data used in this study are sourced from publicly available datasets. |
| Specimen deposition | NA |
| Dating methods | Sample dates were obtained from published materials. |

☐ Tick this box to confirm that the raw and calibrated dates are available in the paper or in Supplementary Information.

| | |
|---|---|
| Ethics oversight | NA |

Note that full information on the approval of the study protocol must also be provided in the manuscript.

