## [Peer Review File · Nature Ecology & Evolution]

Peer Review Information

Journal: Nature Ecology & Evolution

Manuscript Title: Admixture has obscured signals of historical hard sweeps in humans

Corresponding author name(s): Yassine Souilmi, Raymond Tobler, Angad Johar, Christian D. Huber

Editorial Notes:

Reviewer Comments & Decisions:

Decision Letter, initial version:
--

19th January 2022

Dear Dr Souilmi,

Your Article entitled "Admixture has obscured signals of historical hard sweeps in humans" has now been seen by three reviewers, whose comments are attached. In the light of their advice, we have decided that we cannot offer to publish your manuscript in Nature Ecology & Evolution.

From the reports, you will see that while they find your work of some potential interest, the reviewers raise concerns about the advance your findings represent over earlier work and the strength of the novel conclusions that can be drawn at this stage. We feel that these criticisms are sufficiently important as to preclude publication of your work in Nature Ecology & Evolution.

I am sorry that we cannot be more positive on this occasion, but hope that you find the reviewers' comments helpful when preparing your paper for resubmission elsewhere.

[REDACTED]

Reviewer expertise:

Reviewer #1: human population genetics, detecting hard and soft sweeps

Reviewer #2: allele fixation in human population genetics

Reviewer #3: human population genetics (including ancient genomes)

Reviewers Comments:

Reviewer #1 (Remarks to the Author):

Summary

In this paper the authors argue that admixture is an important confounding factor in studies of positive selection and has likely masked or distorted the signals of historical hard sweeps in previous selection studies. The authors analyze ancient and modern human genomes with a method they previously developed called SweepFinder 2, which they show is robust to demographic bottlenecks, missing data, ascertainment bias and alignment errors. This work adds to the growing literature on cautioning the interpretation of tempo and mode in evolution in natural populations with complex demography such as admixture.

Critiques

1. The claims of this paper that “The extent of past hard sweep events have likely been underestimated in natural populations in general, biasing our understanding of the mode and tempo of adaptation in humans and other species” seem grander than is warranted for several reasons.

The authors seem to want to cast critique on over-interpreting the tempo and mode of evolution in natural populations. In the introduction and last paragraph of their paper, they cite a recent paper (Harris et al. 2018, PLoS Genetics) that was extremely problematic for the field. Two rebuttals were written to this paper (Garud et al. 2021 and Feder et al. 2021), but neither are cited in this manuscript. In particular, Garud et al. 2021 addresses the many misleading claims made by Harris et al. 2018 regarding the potential confounding nature of admixture in classifying hard sweeps as soft. If this paper is going to frame its importance around Harris et al. 2018, then it is important to also cite Garud et al. 2021 and explain if this paper disagrees with the conclusions of Garud et al. 2021 and why. Otherwise, unfortunately, I think that this is a one sided review of the literature that further perpetuates damaging and unsubstantiated claims in the field.

Nonetheless, I do not think that this paper is at odds with the recent literature claiming that soft sweeps are common in the recent past in natural populations (e.g. Schrider and Kern 2018, Garud 2015, Sheehan and Song 2015, + the entire body of work reviewing the prevalence of soft sweeps in Messer and Petrov 2013). In fact, hard sweeps could have been common in the past and soft sweeps could have been common in contemporary populations. Thus, acknowledging that both can be a possibility is a fairer treatment of the literature.

That being said, the present work itself is a lopsided investigation into the prevalence of hard sweeps

2in the past. The authors use SweepFinder2 to identify hard sweeps, but fail to acknowledge that in their own paper, Huber et al 2015, that the same statistic is capable of detecting soft sweeps. Specifically, in Huber et al. 2015, the authors write: "The gene is also an outlier for haplotype-based sweep statistics for detecting incomplete soft or hard sweeps, in an African population (Ferrer-Admetlla et al. 2014)." Thus, how can we conclude that these sweeps that the authors have detected are hard to begin with?

Admittedly, the authors' statistics should be less sensitive to soft sweeps than hard sweeps, as Pennings and Hermission showed in Soft Sweeps 3 that LD-based statistics are more powerful than SFS based statistics. Thus, while the authors would like to state that the tempo and mode of evolution may have been misinterpreted in the context of admixture, it is unclear how common soft sweeps were in the past without a scan to demonstrate it. Hard sweeps could have been common, but soft sweeps could have been common too.

There is some mention of attempting to discern the probability that the sweeps detected arose from SGV versus de novo mutation in Figure 6. Here the authors examine the probability that the sweep arose from SGV and find that both de novo mutation and SGV could have non-negligible probabilities of giving rise to the sweeps detected. Thus, I am not sure what the motivation or conclusion is for this analysis. Is it to demonstrate that sweeps in the past were more likely to have been hard (which the simulations don't support) or justification for only looking for hard sweeps using SF2 (which is poor)?

2. The authors mention that reduction of hard sweep signal cannot be explained by the degradation of the sweep through drift. They test if sweep signal is dependent on sweep antiquity and find that local sweeps are more susceptible to post admixture degradation while sweeps closer to out of Africa events (older) are more robust to population admixture (fig s17). Admixture may be one important factor decreasing the signal of selective sweeps, but how does this scenario compare to the case of non-admixture, where other factors such as recombination and de novo mutations reduce the sweep signal.

3. Figure 2B seems to contain important and relevant information, but it is glossed over in the main text. Specifically, this figure seems to be fairly important in demonstrating the ability to detect hard sweeps originating before and after admixture. First, why are we even considering $Q \sim 0.1$? Do the authors believe in these results? It seems like sweeps were called with $Q = 0.01$, so why consider other Q s? What is the reader supposed to take from this? Second, if using a reasonable Q value of 0.01, then the postadmixed seem to do equally well if not outperform the admixed populations. Thus, I do not understand why the authors conclude that they cannot detect hard sweeps that originated pre-admixture.

4. The authors classify the age of the sweeps by assuming that the onset of selection was at least as old as the oldest sample exhibiting the beneficial haplotype. Using this classification, they detect 44/57 sweep haplotypes are present by ~ 35 ka and suggest that the onset of selection for these sweeps arose around the time of the out of Africa event. However they only detect four sweeps at ~ 30 ka. Why is this enough to suggest that local sweeps should be more susceptible to post-admixture signal degradation?

Additionally, in figure 5: The authors do not seem to explain the poor power in the steppe populations – only WGS.

5. In general, I find the presentation of the figures in this paper to be confusing. First, in Figures 2 and 4, it is difficult to keep track of which populations are admixed and which ones are pre-admixed. Could the authors delineate (and perhaps even group) the populations as such on their figures? E.g. Anatolia, Steppe, and WHG are preadmixed and EF, LNBA, modern Europeans are post admixed? It is very hard to keep referencing Figs 1 and 2A, where not all the populations are indicated in the first place.

Minor:

1. More on Figure 4:

- I cannot keep straight which populations I should look at for sweeps starting within the last 35K yrs. How do I match Fig 4A with the x-axis of Fig. 4B?
- Figure 4C, unclear how to match the sweep numbers with that of Fig 4A

2. Figure 5: panel C is referenced but is missing.

3. Cite DeGiorgio 2016 for SweepFinder2?

4. Clarify that the following statement is with regards to classification of the age of a sweep: “These results demonstrate that admixture can sufficiently distort the genetic signals resulting from a hard sweep, leading to the misclassification of the inferred mode of selection in studies where it is not explicitly accounted for.”

Reviewer #2 (Remarks to the Author):

In this manuscript, the authors evaluate how admixture events can affect the power of detection of hard sweeps and identification of mode of positive selection (i.e., complete vs partial sweeps). This study is conducted in humans, using ancient and some modern human populations. Population genetic modeling using forward simulations as well as analytical methods are used to understand these questions, accounting for the demographic history of these populations. Overall, I really like the study- it's rigorous and will be very valuable for the community that works on humans as well as for population geneticists in general. I do however feel that there are 2 fronts at which it can improve considerably:

1. Currently the paper is written assuming that one knows everything about human populations and that can be an impediment for researchers who are not familiar with these populations.
2. The text and figure legends need to be much clearer about what exactly was done. In the present state, it is sometimes very hard to understand what you exactly did. And there are no line numbers,

4which makes it difficult to give comments.
Following are my specific comments.

Major Comments:

1. While performing the forward simulations in SLiM, did you account for the specific recombination rate in those regions of the 57 sweeps? At least broad scale rates should be the same across populations, right? That would make the analysis more robust.
2. "and grouped these samples into 18 distinct ancient populations..." -> Fig1 is nice but it's really difficult to understand the actual populations and how they connect to Figure 2. For instance, which of those 18 populations are before and after the Holocene admixture events? A much more elaborate figure that shows these populations on a plot of the demographic history will be extremely helpful to the readers. It would also be really great if various events are marked on it, for instance, the out of Africa events.
3. My reading of the Figure is that the trough in diversity coincides with the decrease in recombination rates, so how is that an evidence of a sweep? It seems exactly the opposite, in fact.
4. Page 7, "We first investigated a model where selection is active ..." -> Can you justify a change in selection pressure coincident with an admixture event a bit more? I'm a bit unsure about why that is. Are there some other citations? Also, it would be great to have another figure that shows where this change in "s" is being modeled.

Minor Comments:

1. Page 3 -> "Importantly, testing our sweep detection pipeline on simulated..." -> I believe you did not estimate this demographic history and are using it from another manuscript, right? If yes, please mention it here clearly. In general it was unclear how you obtained it.
2. Page 5, "taken together, the evidence strongly implies that MHC-III...likely gone unnoticed..." -> I think that is a bit strong. Hasn't it gone unnoticed because no one has looked at these ancient populations?
3. Page 9, "AMH occupation" -> what is AMH?
4. I like the part about looking at the probability of sweeps due to SGV vs de novo mutation. Are these calculated assuming demographic equilibrium? If yes, that caveat should be mentioned.
5. Page 10, "41 sweeps having $s > 1\%$..." -> I was wondering about this the entire paper. It would be much better if this information showed up right at the beginning of the results section.

Reviewer #3 (Remarks to the Author):

The authors pose an interesting question: given the rampant admixture that has occurred in the course of human evolution, now documented in hundreds of studies, could it be that methods to detect hard sweeps based on current day population labels miss many of them? Phrased in another way, if current day population labels actually regroup mixtures of highly diverged ancestries, would we expect to detect a strongly beneficial allele that reached fixation in only one of these ancestries? But then to address this question, the authors do something quite odd in my view, which is to consider that one step back, populations suddenly have a meaning again, and admixture is not an issue. Specifically, they consider the three ancestries that gave rise to modern Europeans, and imagine those to have persisted long enough for beneficial alleles to ascend from rare to fixation. Yet we know that

5those ancestries too are mixtures of other ancestries. (Contrary to what is stated in the SOM, them forming a blob in a PCA plot is not evidence for them representing an enduring un-admixed population—so do Europeans in the 1000G data for example.) So the whole set up of this study seems inconsistent to me.

In a sense, the question that the authors raise is in my view deeper than what they contend with here: given that mixing of divergent groups was the norm throughout human evolution, likely at rates higher than the time scale it takes for a rare mutation to reach fixation (~50,000 years?), are hard sweeps even the right model for the behavior of strongly beneficial alleles? By using the hard sweep model, could we be missing quasi-Mendelian adaptations?

Instead, this study just kicks the can down the road, by considering DNA samples from the three labels that mixed to form present day Europeans as populations, and looking for sweeps in them instead. So then we have to ask ourselves how errors in the specification of the demographic model for the “ancestral populations” that gave rise to current day Europeans could lead to false positives in their sweep detection. Given that mis-specification of the demographic history is a huge issue in that regard, some sanity checks are in order. (To be clear, the authors present many analyses of possible artifacts, but none on what seems likely to be the thorniest issue.) Likewise, the lessons of their Figures 4-6 are dependent on the specifics of the simple demographic model they choose, so it would be good to know how much independent support there is for their model and how well it fits data at neutral sites.

In that regard, it seems a weird omission that the authors don't tell us about 56 of the 57 sweep signals not in the MHC, other than that they have higher F_{st} to Yoruba: where are they? are they plausible candidates etc?

Finally, the section on the mutational basis of sweeps is missing previous references about what happens when previously deleterious mutations become strongly favored, and how these are not distinguishable from hard sweeps, as well as the interaction with demography, such as Orr & Betancourt 2001 Genetics and Teshima et al. 2006 Genome Research, among others.

**Although we cannot publish your paper, it may be appropriate for another journal in the Nature Portfolio. If you wish to explore the journals and transfer your manuscript please use our <https://mts-natecolevol.nature.com/cgi-bin/main.plex?el=A3Cn2Fuw2A5BQve5X6A9ftddz2rLeWN5mirwFsrXSCxQZ> manuscript transfer portal. If you transfer to Nature journals or the Communications journals, you will not have to re-supply manuscript metadata and files. This link can only be used once and remains active until used.

All Nature Portfolio journals are editorially independent, and the decision on your manuscript will be taken by their editors. For more information, please see our [manuscript transfer FAQ](http://www.nature.com/authors/author_resources/transfer_manuscripts.html?WT.mc_id=EMI_NPG_1511_AUTHORTRANSF&WT.ec_id=AUTHOR) page.

Note that any decision to opt in to In Review at the original journal is not sent to the receiving journal on transfer. You can opt in to *In Review* at receiving journals that support this service by choosing to modify your manuscript on transfer. In Review is available for primary research manuscript types only.

** For Nature Research general information and news for authors, see <http://npg.nature.com/authors>.

Author Rebuttal to Initial commentsReviewer #1 (Remarks to the Author):

Summary

In this paper the authors argue that admixture is an important confounding factor in studies of positive selection and has likely masked or distorted the signals of historical hard sweeps in previous selection studies. The authors analyze ancient and modern human genomes with a method they previously developed called SweepFinder 2, which they show is robust to demographic bottlenecks, missing data, ascertainment bias and alignment errors. This work adds to the growing literature on cautioning the interpretation of tempo and mode in evolution in natural populations with complex demography such as admixture.

Critiques

1. The claims of this paper that “The extent of past hard sweep events have likely been underestimated in natural populations in general, biasing our understanding of the mode and tempo of adaptation in humans and other species” seem grander than is warranted for several reasons.

The authors seem to want to cast critique on over-interpreting the tempo and mode of evolution in natural populations. In the introduction and last paragraph of their paper, they cite a recent paper (Harris et al. 2018, PLoS Genetics) that was extremely problematic for the field. Two rebuttals were written to this paper (Garud et al. 2021 and Feder et al. 2021), but neither are cited in this manuscript. In particular, Garud et al. 2021 addresses the many misleading claims made by Harris et al. 2018 regarding the potential confounding nature of admixture in classifying hard sweeps as soft. If this paper is going to frame its importance around Harris et al. 2018, then it is important to also cite Garud et al. 2021 and explain if this paper disagrees with the conclusions of Garud et al. 2021 and why. Otherwise, unfortunately, I think that this is a one sided review of the literature that further perpetuates damaging and unsubstantiated claims in the field.

The paper by Garud et al. (2021) deals with the claim by Harris et al. (Harris et al. 2018) that soft sweep signatures in North American *Drosophila* that were found in Garud et al. (2015) can be readily explained by confounding effects of complex demographic history, in particular recent admixture. Garud et al. (2021) convincingly show that a neutral model is incompatible with the outliers of haplotype statistics that they find, under a wide range of tested demographic histories. They also simulate selection on top of complex admixture demographics and show that their approach can successfully discriminate soft from hard selective sweeps. However, in the simulations of Garud et al. (2021) the beneficial mutation is introduced only after the admixture event, never before. Thus, they do not deal with the confounding effect that admixture has on selection signals when selection happens before the admixture event, the main focus of

our study. It is in fact not possible to simulate this scenario with the simulation software that they used (msms), as is pointed out in our Supplementary Method (*Coalescent simulation approach and issues*). We thus agree with Garud et al. (2021) that the selection signatures in North American *Drosophila melanogaster* are difficult to explain by a purely neutral model, but we want to point out that correct classification and timing of selection events could still be confounded. In fact, Garud et al. (2015) themselves state in their Discussion that "*A likely possibility is that we observe signatures of multiple local hard sweeps arising within sub-demes of the North American Drosophila population or in the ancestral European and African populations prior to admixture, that combine to generate signatures of soft sweeps*". Similar confounding effects were found in an extensive simulation study by Zheng and Wiehe (2019, PLoS Computational Biology), which came to the conclusion that "*the claim that most sweeps in the evolutionary history of humans were soft, may have to be reconsidered*". By investigating populations both before and after admixture using ancient DNA, importantly, we provide the first empirical evidence that admixture indeed confounds signatures of selection, in particular regarding the inferred spatio-temporal dynamics of adaptation. We have added citations to Garud et al. (2021) and Zheng and Wiehe (2019) to provide a broader and more balanced selection of cited literature in our introduction.

Nonetheless, I do not think that this paper is at odds with the recent literature claiming that soft sweeps are common in the recent past in natural populations (e.g. Schrider and Kern 2018, Garud 2015, Sheehan and Song 2015, + the entire body of work reviewing the prevalence of soft sweeps in Messer and Petrov 2013). In fact, hard sweeps could have been common in the past and soft sweeps could have been common in contemporary populations. Thus, acknowledging that both can be a possibility is a fairer treatment of the literature.

We agree that the results of our study are not at odds with the recent literature on soft sweeps and did not make any statement about the occurrence or frequency of soft selective sweeps in humans in our manuscript. However, we have added a sentence to acknowledge that soft sweep signals could still be a common type of selection in humans in the Discussion:

"While these factors imply that the extent of past hard sweep events has likely been underestimated in natural populations in general, we caution that they do not directly challenge results from previous studies proposing a substantial role for soft sweeps in the adaptive history of humans⁶ and other species⁶⁶. Rather we reiterate the conclusions of other recent work^{67,68} that advocate for improved modeling of complex but widespread evolutionary and demographic processes to achieve an unbiased understanding of the mode and tempo of adaptation in humans and other species."

That being said, the present work itself is a lopsided investigation into the prevalence of hard sweeps in the past. The authors use SweepFinder2 to identify hard sweeps, but fail to acknowledge that in their own paper, Huber et al 2015, that the same statistic is capable of detecting soft sweeps. Specifically, in Huber et al. 2015, the authors write: “The gene is also an outlier for haplotype-based sweep statistics for detecting incomplete soft or hard sweeps, in an African population (Ferrer-Admetlla et al. 2014).” Thus, how can we conclude that these sweeps that the authors have detected are hard to begin with?

The Huber et al. 2015 study reported a significant SweepFinder2 signal for a gene in European data that had been previously detected as an incomplete soft *or* hard sweep signal in an African population in another study. Those two populations had been separated for at least 100,000 years and thus had ample opportunity for independent adaptive events. However, while it is quite likely that selection has caused a soft or incomplete sweep signal in Africa but a fixed hard sweep signal in Europe, this doesn't mean that SweepFinder2 has power to detect soft sweeps. In fact, SFS-based method such as SweepFinder2 have been shown to have basically no power to detect the soft sweep signatures that are detected by haplotype statistics such as nSL (Ferrer-Admetlla et al. 2014), see also Pennings and Hermisson (2006). It can however detect selection from standing variation if the starting frequency of the selected mutation is low (<1%), as we point out in great detail in the Results section “*The mutational basis of the Eurasian hard sweeps*”. Again, this is totally consistent with previous literature, showing that selection from rare standing variation (<1%) can not be differentiated from selection from de novo mutation using population genetic summary statistics (Peter et al. 2012, Nakagome et al. 2019).

Admittedly, the authors' statistics should be less sensitive to soft sweeps than hard sweeps, as Pennings and Hermisson showed in Soft Sweeps 3 that LD-based statistics are more powerful than SFS based statistics. Thus, while the authors would like to state that the tempo and mode of evolution may have been misinterpreted in the context of admixture, it is unclear how common soft sweeps were in the past without a scan to demonstrate it. Hard sweeps could have been common, but soft sweeps could have been common too.

We agree and have changed the text to more explicitly acknowledge that soft sweep signals could still be a common type of selection in humans (see above).

There is some mention of attempting to discern the probability that the sweeps detected arose from SGV versus de novo mutation in Figure 6. Here the authors examine the probability that the sweep arose from SGV and find that both de novo mutation and SGV could have non-negligible probabilities of giving rise to the sweeps detected. Thus, I am not sure what the motivation or conclusion is for this analysis. Is it to

demonstrate that sweeps in the past were more likely to have been hard (which the simulations don't support) or justification for only looking for hard sweeps using SF2 (which is poor)?

This theoretical analysis investigates the mutational basis of the hard sweep signals that we observe in the ancient populations. It suggests that the hard sweep signals might have been generated by either selection on de novo mutations or selection on rare standing genetic variation. Hard sweep signals from standing genetic variation are quite likely when the mutation was slightly deleterious before the environmental change ($s \sim -0.001$), but not if the mutation was very deleterious ($s \sim -0.01$) or neutral ($s \sim 0$). Again, we do not claim that there are no soft sweep signals (multiple sweep haplotypes at the same locus) in the ancient DNA data – this could very well be the case. However, current soft sweep detection methods can't be applied to aDNA data due to its low quality.

2. The authors mention that reduction of hard sweep signal cannot be explained by the degradation of the sweep through drift. They test if sweep signal is dependent on sweep antiquity and find that local sweeps are more susceptible to post admixture degradation while sweeps closer to out of Africa events (older) are more robust to population admixture (fig s17). Admixture may be one important factor decreasing the signal of selective sweeps, but how does this scenario compare to the case of non-admixture, where other factors such as recombination and de novo mutations reduce the sweep signal.

It is known that the sweep signal detected by SweepFinder2 decays after about $0.2 N_e$ generations without admixture (Huber et al. 2016). This result was replicated in the current study (Fig. S15) and is consistent with previous studies of SFS-based sweep statistics (Przeworski 2002, Sabeti et al. 2006). Importantly, this implies that the hard sweep signals in Eurasian human genomes are expected to remain visible to our analytical pipeline for around 70,000 years in the absence of other effects, given estimated effective population sizes (Fig. S15). We have now clarified this in the main text,

"This dramatic reduction in hard sweep signals was not an artifact of differences in either data quality or quantity between modern and ancient populations (Figs. S11-S14; SI Methods). Nor could their absence be explained by the degradation of the sweep signals through random allele frequency changes (i.e. genetic drift) and new mutations, as hard sweep signals in Eurasian human genomes are expected to remain visible to our analytical pipeline for around 70,000 years in the absence of other effects (Fig. S15; see SI Methods and also refs.^{20,24})."

3. Figure 2B seems to contain important and relevant information, but it is glossed over in the main text. Specifically, this figure seems to be fairly important in demonstrating the ability to detect hard sweeps originating before and after admixture. First, why are we even considering $Q \sim 0.1$? Do the authors believe in these results? It seems like sweeps were called with $Q=0.01$, so why consider other Q s? What is the

raeder supposed to take from this? Second, if using a reasonable Q value of 0.01, then the postadmixed seem to do equally well if not outperform the admixed populations. Thus, I do not understand why the authors conclude that they cannot detect hard sweeps that originated pre-admixture.

We agree with the reviewer that the false discovery rate (FDR) plots in Fig. 2B with Q-value cutoffs different than 0.01 (i.e. $Q < 0.05$ and $Q < 0.1$) are not particularly relevant for interpreting our results since we eventually used a Q-value cutoff of 0.01 to call sweep regions. We have now removed these subplots from Fig. 2 to avoid risking confusion.

However, regarding the second point, we want to emphasize that Fig. 2B does not provide any information about power. This plot of FDR only indicates the proportion of the 57 sweep signals in our data that could be explained by a neutral demographic model. These results are not based on simulations of selective sweeps, thus can't provide information about the ability to detect sweeps in pre- vs. post-admixed populations. Instead, Fig 5. shows the low power to detect hard sweeps originating pre-admixture.

4. The authors classify the age of the sweeps by assuming that the onset of selection was at least as old as the oldest sample exhibiting the beneficial haplotype. Using this classification, they detect 44/57 sweep haplotypes are present by ~ 35 ka and suggest that the onset of selection for these sweeps arose around the time of the out of Africa event. However they only detect four sweeps at ~ 30 ka. Why is this enough to suggest that local sweeps should be more susceptible to post-admixture signal degradation?

While there are only four sweeps found in the Vestonice sample but not older specimens (i.e. giving a sweep age of ~ 30 ka), there are an additional 9 sweeps that are absent in specimens older than 19ka. Our statistical testing **suggests that for both groups there is a difference in the proportion of significant sweep signals across ancient populations (logistic regression, $p = 1.28e-03$ and $p = 1.68e-05$).**

Additionally, in figure 5: The authors do not seem to explain the poor power in the steppe populations – only WGS.

The poor power in the Steppe population for sweeps starting at 44 and 36 ka is an artifact of our simulation setup (see Fig. S19). Given our setup, sweeps starting at 44 and 36 ka are simply not shared by the Steppe populations and thus can't be detected there. Note however that there is high power to detect sweeps that start 55 ka since these sweeps are old enough to be shared with and fixed within the sampled Steppe population. We did not include results from "Steppe-specific" sweeps (i.e. sweeps that are introduced into the immediate Steppe-ancestral populations) since these results do not provide any further insight into the effect of admixture on sweep detection.

5. In general, I find the presentation of the figures in this paper to be confusing. First, in Figures 2 and 4, it is difficult to keep track of which populations are admixed and which ones are pre-admixed. Could the authors delineate (and perhaps even group) the populations as such on their figures? E.g. Anatolia, Steppe, and WHG are preadmixed and EF, LNBA, modern Europeans are post admixed? It is very hard to keep referencing Figs 1 and 2A, where not all the populations are indicated in the first place.

We thank the reviewer for this helpful suggestion, and now clearly indicate in Fig. 2 which populations are "Ancestral European", "Admixed (Anatolia, WHG)", and "Admixed (Anatolia, WHG, Steppe)".

Minor:

1. More on Figure 4:

– I cannot keep straight which populations I should look at for sweeps starting within the last 35K yrs. How do I match Fig 4A with the x-axis of Fig. 4B?

Fig. 4A provides an overview of the estimated age of the 57 sweeps, i.e. as indicated by the oldest of the five Upper Paleolithic specimens in which the sweep haplotype can be found. Based on this classification, we have stratified the analysis shown in Fig. 4B on the y-axis. Thus, there is a direct relationship between Fig. 4A and the y-axis of Fig. 4B (but not the x-axis).

– Figure 4C, unclear how to match the sweep numbers with that of Fig 4A

The sweep numbers in Fig. 4C directly match the numbers in Fig. 4A. For example, there are $6+2+2+6=16$ sweeps that have a sweep haplotype that is observed in Ust Ishim, which is also the uppermost number in Fig. 4A of sweeps that started before Ust Ishim was sampled. However, since there is only one sweep that was found in El Miron (but not any of the older specimens), this category was merged with the 8 sweeps that were not found in any of the Upper Paleolithic specimens.

2. Figure 5: panel C is referenced but is missing.

We have fixed this error.

3. Cite DeGiorgio 2016 for SweepFinder2?

We have now added a citation to the application note of Sweepfinder2, i.e. DeGiorgio et al. 2016.

4. Clarify that the following statement is with regards to classification of the age of a sweep: "These results demonstrate that admixture can sufficiently distort the genetic signals resulting from a hard sweep, leading to the misclassification of the inferred mode of selection in studies where it is not explicitly accounted for."

This is a good point. We changed this sentence to "...leading to the misclassification of the inferred spatio-temporal dynamics of adaptation in studies where it is not explicitly accounted for".

Reviewer #2 (Remarks to the Author):

In this manuscript, the authors evaluate how admixture events can affect the power of detection of hard sweeps and identification of mode of positive selection (i.e., complete vs partial sweeps). This study is conducted in humans, using ancient and some modern human populations. Population genetic modeling using forward simulations as well as analytical methods are used to understand these questions, accounting for the demographic history of these populations. Overall, I really like the study- it's rigorous and will be very valuable for the community that works on humans as well as for population geneticists in general. I do however feel that there are 2 fronts at which it can improve considerably:

1. Currently the paper is written assuming that one knows everything about human populations and that can be an impediment for researchers who are not familiar with these populations.
2. The text and figure legends need to be much clearer about what exactly was done. In the present state, it is sometimes very hard to understand what you exactly did. And there are no line numbers, which makes it difficult to give comments.

We thank the Reviewer for the positive comments about the rigor of our study and for pointing out the significance of our results for population genetics. We have now improved the clarity of our manuscript considerably based on the combined Reviewer comments. We have also added line numbers to the revised version of the manuscript.

Following are my specific comments.

Major Comments:

1. While performing the forward simulations in SLiM, did you account for the specific recombination rate in those regions of the 57 sweeps? At least broad scale rates should be the same across populations, right? That would make the analysis more robust.

We have simulated under a genome-wide average recombination rate. We have further evaluated the robustness of our sweep detection pipeline to one order of magnitude larger or smaller recombination rate (Fig. S7). We find that changing recombination rate does not substantially deviate the z-score distribution

from a standard normal distribution. Thus, our approach of normalizing the SweepFinder2 CLR statistic (see Fig. S3) seems to provide ample robustness to varying recombination rates. Further, we don't find a significant relationship between recombination rate and the SweepFinder2 CLR statistic across the genome ($\rho = 0.012$, $p = 0.7$). Thus, we don't suspect recombination rate to be a confounding factor in our study. The full details of this analysis can be found in the Supplemental Material (section *Testing the impact of recombination rate on sweep detection*).

2. “and grouped these samples into 18 distinct ancient populations...” -> Fig1 is nice but it's really difficult to understand the actual populations and how they connect to Figure 2. For instance, which of those 18 populations are before and after the Holocene admixture events? A much more elaborate figure that shows these populations on a plot of the demographic history will be extremely helpful to the readers. It would also be really great if various events are marked on it, for instance, the out of Africa events.

Unfortunately, the exact relationship between the ancestral populations of Europeans is not yet fully resolved, particularly for the Upper Paleolithic (>12,000y). As a result, we prefer not to provide an explicit plot of the demographic history of Europeans. While the key parameters relevant to our study are strongly supported, such as the increase in admixture between highly diverged ancestral populations over the last 10,000 years, we feel that an explicit but necessarily arbitrary plot of demographic history could distract from the main topic of our study. However the suggestion about the need to distinguish the key populations is a good one, and we now provide labeling in Fig. 2 clarifying which populations are pre- versus post-admixture.

3. My reading of the Figure is that the trough in diversity coincides with the decrease in recombination rates, so how is that an evidence of a sweep? It seems exactly the opposite, in fact.

A strong relation between recombination rate and diversity typically indicates the effect of linked selection – the diversity-reducing effect of linked selection is stronger in regions with low recombination rate. We agree with the reviewer that the drop in diversity shown in Fig. 3 could also be the result of non-adaptive processes, i.e. background selection. However, background selection would act similarly across all ancient populations, whereas the strong reduction in diversity in the HLA region is only seen in the Anatolian EF population. It is also not seen in any of the other ancestral populations nor in the heavily admixed Bronze Age populations (Fig. S16). Thus, we argue that a local selective sweep in Anatolia is the more likely explanation for why recombination rate is correlated with diversity in this region. We present this argument now in greater detail in a separate supplementary text (Text S3, *HLA locus sweep in Anatolian Neolithic samples*).

4. Page 7, “We first investigated a model where selection is active ... “ -> Can you justify a change in selection pressure coincident with an admixture event a bit more? I’m a bit unsure about why that is. Are there some other citations? Also, it would be great to have another figure that shows where this change in “s” is being modeled.

In these simulations, we aimed to investigate what happens to selection signatures if a mutation sweeps to fixation in a subpopulation, but does not fix again after admixture with another population that did not have the mutation. Thus, we turned off selective pressure after admixture. We agree that an admixture event is not necessarily directly related or linked to a change of selection pressure in human history. However, there is recent evidence that selection pressure on certain mutations or genomic segments has not been constant throughout time but differed during different time periods and in different populations (e.g. Mathieson and Mathieson 2018, Zhang et al. 2020, Jagoda et al. 2018, Yair et al. 2021, Souilmi et al. 2021). Thus, it may be that changes in selection pressure throughout time, or population-specific selection pressures, are more common than previously thought. We have added citations to this literature.

Minor Comments:

1. Page 3 -> “Importantly, testing our sweep detection pipeline on simulated...” -> I believe you did not estimate this demographic history and are using it from another manuscript, right? If yes, please mention it here clearly. In general it was unclear how you obtained it.

This is correct, we used demographic model parameters from Kamm et al. (2019). The demographic history of the Steppe population was further augmented by parameters from two other papers (Damgaard et al. 2018, Jones et al. 2015). This is now clearly stated in the figure caption of Fig. 2A.

2. Page 5, “taken together, the evidence strongly implies that MHC-III...likely gone unnoticed...” -> I think that is a bit strong. Hasn’t it gone unnoticed because no one has looked at these ancient populations?

We thank the reviewer for pointing this out. We have changed the second part of this sentence to say

“Taken together, the evidence strongly implies that the MHC-III region was a target of strong positive selection in the Anatolian EF population and that the underlying hard sweep signal became masked in descendant populations by Holocene-era admixture involving genetically diverged populations. ”

3. Page 9, “AMH occupation” -> what is AMH?

AMH is the standard abbreviation for Anatomically Modern Humans, and we should have explained this. We have included the full name now.

4. I like the part about looking at the probability of sweeps due to SGV vs de novo mutation. Are these calculated assuming demographic equilibrium? If yes, that caveat should be mentioned.

Thanks! In fact, we do not assume equilibrium in these calculations – we assume a 10-fold reduction in population size at the time of the environmental change, reflecting the Out-of-Africa bottleneck. Results for less severe bottleneck scenarios are shown in Fig. S21. We now clarify this in the main text and in the caption for Fig. 6.

5. Page 10, “41 sweeps having $s > 1\%$...” -> I was wondering about this the entire paper. It would be much better if this information showed up right at the beginning of the results section.

We thank the Reviewer for pointing this out, and have added this information to the beginning of the results section.

Reviewer #3 (Remarks to the Author):

The authors pose an interesting question: given the rampant admixture that has occurred in the course of human evolution, now documented in hundreds of studies, could it be that methods to detect hard sweeps based on current day population labels miss many of them? Phrased in another way, if current day population labels actually regroup mixtures of highly diverged ancestries, would we expect to detect a strongly beneficial allele that reached fixation in only one of these ancestries? But then to address this question, the authors do something quite odd in my view, which is to consider that one step back, populations suddenly have a meaning again, and admixture is not an issue. Specifically, they consider the three ancestries that gave rise to modern Europeans, and imagine those to have persisted long enough for beneficial alleles to ascend from rare to fixation. Yet we know that those ancestries too are mixtures of other ancestries. (Contrary to what is stated in the SOM, them forming a blob in a PCA plot is not evidence for them representing an enduring un-admixed population—so do Europeans in the 1000G data for example.) So the whole set up of this study seems inconsistent to me.

In a sense, the question that the authors raise is in my view deeper than what they contend with here: given that mixing of divergent groups was the norm throughout human evolution, likely at rates higher than the time scale it takes for a rare mutation to reach fixation (~50,000 years?), are hard sweeps even the right model for the behavior of strongly beneficial alleles? By using the hard sweep model, could we be missing quasi-Mendelian adaptations?

We thank Reviewer 3 for their interesting and insightful perspective on our study. We totally agree that our results point to a deeper problem of selection signatures in species where admixture is rampant. As a result,

we think our study is very relevant not only for humans but also for many other species. Population structure and historical admixture have long been ignored in studies of genetic adaptation signals – possibly because we previously didn't have ancient DNA that could record the complexities of past demography.

We also agree that the three European ancestral populations (WHG, Steppe, Anatolian EF) could be admixed themselves and that we might miss selection events because of this admixture. However, there are three arguments for why we think that there was more opportunity for strongly beneficial alleles to reach fixation in those ancient ancestral populations than in the more recent populations of the late Holocene:

1) There was a dramatic increase in population mixture during the Holocene, as illustrated by the striking reduction in F_{ST} between populations over the last 10,000 years (a 10-fold reduction in F_{ST} between Eurasian populations, see Lazaridis et al. 2016). Thus, the influence of admixture in modifying sweep signals was arguably considerably more severe in the Holocene than in the Upper Paleolithic. Also, qpAdm models from Lazaridis et al. 2016 as well as Mathieson et al. 2018 suggest that Anatolia, WHG and Steppe provide the predominant sources of ancestry for other Eurasian populations (such as Central and Western European Early farmers and LNBA groups etc) included in our study. There is no evidence suggesting that the converse qpAdm models (i.e. when attempting to represent the ancestry of Anatolia and WHG using the other European early farmers, late farmers and LNBA groups) are true.

2) Empirically, we find more hard sweep signals in older, less admixed populations than in the more admixed Bronze Age and modern populations. This suggests that the accompanying increase in admixture has affected the sweep signals since drift and recombination can not reduce sweep signals in this short amount of time (see also our response to Reviewer 1).

3) As we show both by theoretical calculations and simulations (Fig. S18), even relatively weakly beneficial mutations take less than 50,000 years to reach complete or almost complete fixation. For example, if the mutation has a selection coefficient of 2% and the selection pressure starts when the mutation is a rare standing variant (frequency 0.1%), then the mutation reaches a frequency of >90% in only about 15,000 years on average (Fig. S18). Thus, we think that there is a strong chance that the rate of fixation of sweeps is higher during the Upper Paleolithic than the rate of admixture.

Instead, this study just kicks the can down the road, by considering DNA samples from the three labels that mixed to form present day Europeans as populations, and looking for sweeps in them instead. So then we have to ask ourselves how errors in the specification of the demographic model for the “ancestral populations” that gave rise to current day Europeans could lead to false positives in their sweep detection. Given that mis-specification of the demographic history is a huge issue in that regard, some sanity checks are in order. (To be clear, the authors present many analyses of possible artifacts, but none on what seems

likely to be the thorniest issue.) Likewise, the lessons of their Figures 4-6 are dependent on the specifics of the simple demographic model they choose, so it would be good to know how much independent support there is for their model and how well it fits data at neutral sites.

As the Reviewer has pointed out, we have put a lot of effort into providing strong evidence for selective sweeps that is robust to many different kinds of possible artifacts. We agree that misspecification of the demographic history is an important issue, but we disagree that we haven't accounted for it. In Fig. 2 we show that the false discovery rate of our detection pipeline is robust to single strong bottlenecks associated with the founding Eurasian and subsequent WHG populations (successive 12-fold and 6-fold population size reductions, respectively). Further, we test the influence of a range of admixture proportions (33%, 40%, 50%). In all of these cases, the FDR is not substantially inflated, and the p-value distribution of our test is uniformly distributed under neutral simulations (Fig. S4). Thus, these analyses demonstrate that the distribution of transformed CLR statistics used to detect sweeps closely match distributions under neutral simulations.

In sum, our conservative approach of robustly transforming the per-gene SweepFinder statistic into a normally distributed test statistic (Fig. S3) is a novel approach that controls for the major demographic influences of recent human history. We further note that this approach is more stringent than previous studies that did not take historical population structure of European populations into account (e.g., Schrider and Kern 2017).

In that regard, it seems a weird omission that the authors don't tell us about 56 of the 57 sweep signals not in the MHC, other than that they have higher F_{st} to Yoruba: where are they? are they plausible candidates etc?

As noted in the manuscript, a large proportion of our 57 sweep regions (70%) show significant signals of partial sweeps in previous studies of modern European populations (Johnson & Voight 2018, Pickrell et al. 2009). Further, previously well-established candidate genes of strong adaptive evolution in Europeans are also part of our 57 sweeps (i.e. SLC24A5, MLPH, EDAR, ATXN2, HLA). Thus, our candidate regions are highly consistent with previous literature, although we differ in the interpretation of the spatio-temporal context of selection.

All the genes underlying the 57 sweep regions can be found in Table S2. An exhaustive functional interpretation of the genes is out of the scope of this study and will be published in a separate companion paper.

Finally, the section on the mutational basis of sweeps is missing previous references about what happens when previously deleterious mutations become strongly favored, and how these are not distinguishable from hard sweeps, as well as the interaction with demography, such as Orr & Betancourt 2001 Genetics and Teshima et al. 2006 Genome Research, among others.

We have added respective references to previous literature on selection from standing variation.Decision Letter, first revision:

27th May 2022

Dear Dr Souilmi,

Your manuscript entitled "Admixture has obscured signals of historical hard sweeps in humans" has now been seen by two of the original reviewers, whose comments are attached. Reviewer 3 from the previous round was unable to re-review, but reviewer 1 was able to provide comments on their report, which we have attached below--please ensure that outstanding points are addressed. The reviewers have raised a number of concerns which will need to be addressed before we can offer publication in Nature Ecology & Evolution. We will therefore need to see your responses to the criticisms raised and to some editorial concerns, along with a revised manuscript, before we can reach a final decision regarding publication.

We therefore invite you to revise your manuscript taking into account all reviewer and editor comments. Please highlight all changes in the manuscript text file [OPTIONAL: in Microsoft Word format].

* If you have not done so already please begin to revise your manuscript so that it conforms to our Article format instructions at <http://www.nature.com/natecolevol/info/final-submission>. Refer also to any guidelines provided in this letter.

[REDACTED]

Note: This URL links to your confidential home page and associated information about manuscripts you may have submitted, or that you are reviewing for us. If you wish to forward

21this email to co-authors, please delete the link to your homepage.

Nature Ecology & Evolution is committed to improving transparency in authorship. As part of our efforts in this direction, we are now requesting that all authors identified as 'corresponding author' on published papers create and link their Open Researcher and Contributor Identifier (ORCID) with their account on the Manuscript Tracking System (MTS), prior to acceptance. ORCID helps the scientific community achieve unambiguous attribution of all scholarly contributions. You can create and link your ORCID from the home page of the MTS by clicking on 'Modify my Springer Nature account'. For more information please visit www.springernature.com/orcid.

[REDACTED]

Reviewer expertise:

Reviewer #1: as before

Reviewer #2: as before

Reviewers' comments:

Reviewer #1 (Remarks to the Author):

We thank the authors for their careful consideration of our points raised in the previous round. However, we still have several concerns, which should be addressed in full before publication.

Summary:

The authors evaluate how Admixture could have masked hard sweep signatures, making them undetectable in modern data sets. This is an important question to study as it is important to be cautious of overinterpretation of the tempo and mode of adaptation in natural populations where complex demography, such as admixture, is involved. Through simulations and an analysis of ancient and modern genomes, the authors show convincing evidence that supports the claim that admixture has masked hard sweeps in modern data. However, throughout the paper, the authors mention that

22admixture also leads to sweep misclassification but it's unclear what they mean by this. Moreover, some of their figures continue to be very hard to interpret, especially figure 4. These points are further discussed below.

Critiques

- In line 23 the authors write that admixture can “either mask these signals or lead to erroneous inferences about the underlying modes of selection”, however it’s never clear what they mean by erroneous inference/misclassification. Particularly, in the section “Admixture can lead to misclassification of historical hard sweeps” and from the analysis of figure 4c the authors conclude that “admixture can sufficiently distort the genetic signals resulting from a hard sweep, leading to the misclassification of the inferred spatio-temporal dynamics of adaptation in studies where it is not explicitly accounted for”. In this section, it seems that by misclassification the authors mean complete vs partial sweeps but still hard sweeps. A sweep can be hard and partial, so if a sweep is being detected as a hard sweep pre and post admixture I don’t see why we would say it has been misclassified. Moreover, none of their simulations look at whether admixture results in a higher proportion of sweeps that are misclassified (i.e. incorrect mode of selection instead of not detected). Overall, I don’t think this section is showing that ancient hard sweeps are misclassified but rather that admixture makes them undetectable in modern samples.
- Figures 3A and 4 are still very hard to interpret. It seems that the labels are written assuming the reader is very familiar with the human populations used in the study.
 - For figure 4b, a clearer division of the X axis into pre and post admixed populations would make the figure easier to read.
 - It would be helpful if the order of the panels in 4b and 4c matched the order shown in 4a.
 - In 3A, it’d be helpful to have corresponding years/epochs of the populations shown in each panel.

Minor comments

- Line 130- Since 2 of the sweeps have been detected in modern samples, shouldn’t it be masking 55 and not 57 sweeps?
- Lines 251-252: “which has led to these signals being missed or misinterpreted as other modes of selection in previous studies”. In which studies have these signals been misinterpreted? Add citations.

Reviewer #2 (Remarks to the Author):

The authors have done a good job of addressing my comments and concerns.

Reviewer #1's comments on R#3's initial report

Reviewer 3:

R3: The authors pose an interesting question: given the rampant admixture that has occurred in the course of human evolution, now documented in hundreds of studies, could it be that methods to detect hard sweeps based on current day population labels miss many of them? Phrased in another way, if current day population labels actually regroup mixtures of highly diverged ancestries, would we expect to detect a strongly beneficial allele that reached fixation in only one of these ancestries? But then to address this question, the authors do something quite odd in my view, which is to consider

23that one step back, populations suddenly have a meaning again, and admixture is not an issue. Specifically, they consider the three ancestries that gave rise to modern Europeans, and imagine those to have persisted long enough for beneficial alleles to ascend from rare to fixation. Yet we know that those ancestries too are mixtures of other ancestries. (Contrary to what is stated in the SOM, forming a blob in a PCA plot is not evidence for them representing an enduring un-admixed population—so do Europeans in the 1000G data for example.) So the whole setup of this study seems inconsistent to me.

In a sense, the question that the authors raise is in my view deeper than what they contend with here: given that mixing of divergent groups was the norm throughout human evolution, likely at rates higher than the time scale it takes for a rare mutation to reach fixation (~50,000 years?), are hard sweeps even the right model for the behavior of strongly beneficial alleles? By using the hard sweep model, could we be missing quasi-Mendelian adaptations?

R1: The authors respond to this point by acknowledging that the three ancestral populations could also be admixed but give arguments to support that there was more opportunity for strongly beneficial alleles to reach fixation in these ancestral populations than in more modern populations. They also cite Lazaridis et al. 2016 and Mathieson et al. 2018 as evidence that Anatolia, WHG and Steppe provide the predominant sources for Eurasian populations included in their study. I find their arguments convincing but the authors should include these citations in line 41 and mention that the ancestral populations could also be admixed. Additionally, the authors could acknowledge that their demographic model might not be accurately capturing the true complexity of the population's demographic history?

R3: Instead, this study just kicks the can down the road, by considering DNA samples from the three labels that mixed to form present day Europeans as populations, and looking for sweeps in them instead. So then we have to ask ourselves how errors in the specification of the demographic model for the "ancestral populations" that gave rise to current day Europeans could lead to false positives in their sweep detection. Given that mis-specification of the demographic history is a huge issue in that regard, some sanity checks are in order. (To be clear, the authors present many analyses of possible artifacts, but none on what seems likely to be the thorniest issue.) Likewise, the lessons of their Figures 4-6 are dependent on the specifics of the simple demographic model they choose, so it would be good to know how much independent support there is for their model and how well it fits data at neutral sites.

R1: I agree with the authors' response that shows that they do account for demographic history misspecification by testing different bottleneck strengths and admixture proportions.

R3: In that regard, it seems a weird omission that the authors don't tell us about 56 of the 57 sweep signals not in the MHC, other than that they have higher F_{st} to Yoruba: where are they? are they plausible candidates etc?

R1: In the response to Reviewer 3, the authors write that a large proportion of their 57 sweep regions appear in previous studies of modern European populations, including some well-established genes of strong adaptive evolution such as SLC24A5, MLPH, HLA and others. This is confusing since in line 82 they mention that only two of the 57 SF2 outliers are detected in modern populations and have been reported in previous studies. Then, in line 156 they mention that ~66% of the 57 sweeps appear as

24partial sweeps in prior studies of positive selection in modern European populations. In their response, the authors write that “our candidate regions are highly consistent with previous literature, although we differ in the interpretation of the spatio-temporal context of selection”. What do they mean by this spatio-temporal interpretation? Is it that the sweeps appear as partial and not complete sweeps in modern populations? If sweeps are hard pre and post admixture I don’t think they are being misinterpreted. The authors should be clearer on which of their 57 sweep regions overlap with previous literature and clarify what they mean by not agreeing with the “spatio-temporal context of selection” of these previously identified sweeps.

Finally, the section on the mutational basis of sweeps is missing previous references about what happens when previously deleterious mutations become strongly favored, and how these are not distinguishable from hard sweeps, as well as the interaction with demography, such as Orr & Betancourt 2001 Genetics and Teshima et al. 2006 Genome Research, among others.

R3: The authors added citations to previous literature on selection from the standing variation.

*****END*****

Author Rebuttal, first revision:Reviewer #1 (Remarks to the Author):

We thank the authors for their careful consideration of our points raised in the previous round. However, we still have several concerns, which should be addressed in full before publication.

Summary:

The authors evaluate how Admixture could have masked hard sweep signatures, making them undetectable in modern data sets. This is an important question to study as it is important to be cautious of overinterpretation of the tempo and mode of adaptation in natural populations where complex demography, such as admixture, is involved. Through simulations and an analysis of ancient and modern genomes, the authors show convincing evidence that supports the claim that admixture has masked hard sweeps in modern data. However, throughout the paper, the authors mention that admixture also leads to sweep misclassification but it's unclear what they mean by this. Moreover, some of their figures continue to be very hard to interpret, especially figure 4. These points are further discussed below.

We thank Reviewer 1 for their careful evaluation, and for the helpful remarks that have allowed us to clarify both text and figures. We particularly appreciate the positive comments regarding the importance of the study and the convincing level of evidence presented.

Critiques

- In line 23 the authors write that admixture can “either mask these signals or lead to erroneous inferences about the underlying modes of selection”, however it's never clear what they mean by erroneous inference/misclassification. Particularly, in the section “Admixture can lead to misclassification of historical hard sweeps” and from the analysis of figure 4c the authors conclude that “admixture can sufficiently distort the genetic signals resulting from a hard sweep, leading to the misclassification of the inferred spatio-temporal dynamics of adaptation in studies where it is not explicitly accounted for”. In this section, it seems that by misclassification the authors mean complete vs partial sweeps but still hard sweeps. A sweep can be hard and partial, so if a sweep is being detected as a hard sweep pre and post admixture I don't see why we would say it has been misclassified. Moreover, none of their simulations look at whether admixture results in a higher proportion of sweeps that are misclassified (i.e. incorrect mode of selection instead of not detected). Overall, I don't think this section is showing that ancient hard sweeps are misclassified but rather that admixture makes them undetectable in modern samples.

We thank Reviewer 1 for pointing out the potential confusion around this terminology. We agree that our study is not so much concerned with misclassification (i.e., into hard vs. soft sweeps) but rather with the masking of hard sweep signals in modern data by admixed non-sweep haplotypes. Accordingly, we have changed the respective section heading to “Admixture can obscure historical hard sweeps”. We have further changed the concluding sentence of this section to:

"These results demonstrate that admixture can sufficiently distort the genetic signals resulting from ancient fixed sweeps, often leading to haplotype patterns in admixed populations that are misinterpreted as resulting from recent and potentially ongoing selection (Günther and Schmid 2011; Ronen et al. 2015; Sabeti 2006; Voight et al. 2006; Johnson and Voight 2018)."

The text in line 23, however, seems accurate to us. It reflects published simulation results that show that incorrect modelling of population structure and migration/admixture can lead to incorrect *classification* of the mode of selection, i.e. adaptation from standing variation versus de novo mutation (e.g., Zheng and Wiehe 2019), and potential masking of selection events (e.g., Huber et al. 2014).

- Figures 3A and 4 are still very hard to interpret. It seems that the labels are written assuming the reader is very familiar with the human populations used in the study.
 - For figure 4b, a clearer division of the X axis into pre and post admixed populations would make the figure easier to read.
 - It would be helpful if the order of the panels in 4b and 4c matched the order shown in 4a.
 - In 3A, it'd be helpful to have corresponding years/epochs of the populations shown in each panel.

We have modified Fig. 3 and Fig. 4 accordingly. First, we have added the years/epochs of the populations shown in Fig. 3A. Second, we clearly indicate the pre- and post-admixed populations in Fig. 4b by colour coding into the groups that were designated in Fig. 2A (i.e., Ancestral European; Admixed (Anatolia, WHG); Admixed (Anatolia, WHG, Steppe)). Finally, we have matched the order in 4b and 4c with the order shown in 4a.

Fig. 3

Fig. 4

Minor comments

- Line 130- Since 2 of the sweeps have been detected in modern samples, shouldn't it be masking 55 and not 57 sweeps?

This is correct, thanks. Since we want to make a more general statement here, we simply removed the reference to the 57 sweeps: "The Anatolian EF-specific MHC-III hard sweep suggests that if Holocene admixture is responsible more generally for masking hard sweep signals in modern European genomes, then...".

- Lines 251-252: "which has led to these signals being missed or misinterpreted as other modes of selection in previous studies". In which studies have these signals been misinterpreted? Add citations.

A large proportion (70%) of the 57 sweeps appear as partial sweeps in prior studies of positive selection in modern European populations using haplotype-length based methods (*iHS*, *XP-EHH*; Fig. 4C). First, we note that these studies follow an empirical outlier approach (e.g. taking the 1% genome-wide most extreme windows in Johnson and Voight 2018 and Pickrell et al. 2009) and thus can not be considered as rigorous as our study where we control the false discovery rate based on realistic demographic simulations.

Further, we argue that haplotype-based selection signals at our 57 ancient sweeps lead to misinterpretation in the following sense:

- Haplotype-based signals such as *iHS* are interpreted as recent sweeps (e.g., < 30,000 years old in Sabeti et al. 2006; "mainly Holocene era" in Voight et al. 2006), because long-range haplotypes of a partial sweep persist for relatively short periods of time since recombination rapidly breaks down the haplotype (Sabeti 2006). Our analysis, however, suggests that a majority of the sweeps (84%) are older than 30,000 years and thus much older than the haplotype-based signal would suggest.

2. Haplotype-length based signals are typically interpreted as ongoing sweeps at still evolving genes (e.g., Günther and Schmid 2011; Ronen et al. 2015; Sabeti 2006; Voight et al. 2006; Johnson and Voight 2018). Our study suggests that haplotype-based partial sweep signals could alternatively be the result of ancient fixed sweeps subjected to older, transitory selection pressures.

We now clarify the sentence in lines 255-264 in the Discussion to say:

"Our empirical and simulation results implicate Holocene-era admixture as the primary factor attenuating these historical sweep signals, which has led to them being missed in previous studies, or detected instead as empirical genome-wide outliers of haplotype-based selection statistics (Pickrell et al. 2009; Johnson and Voight 2018). Such haplotype-based outliers are typically interpreted as ongoing partial sweeps resulting from recent selection (Sabeti 2006; Johnson and Voight 2018; Günther and Schmid 2011; Ronen et al. 2015; Voight et al. 2006), as the underlying haplotype patterns decay quickly over time and become largely undetectable for selection starting more than 30,000 years ago in humans (Sabeti 2006; Voight et al. 2006). However, our analyses suggest that most (85%) of the 41 sweeps that overlap with a haplotype-based outlier were already under selection by 30ka, implying that these signals more likely result from admixture-driven dilution of hard sweeps that mostly began before 30,000 years ago."

Reviewer #2 (Remarks to the Author):

The authors have done a good job of addressing my comments and concerns.

We thank Reviewer 2 for their advice in the previous round of comments, which significantly improved the quality and clarity of our manuscript.

Reviewer #1's comments on R#3's initial report

Reviewer 3:

R3: The authors pose an interesting question: given the rampant admixture that has occurred in the course of human evolution, now documented in hundreds of studies, could it be that methods to detect hard sweeps based on current day population labels miss many of them? Phrased in another way, if current day population labels actually regroup mixtures of highly diverged ancestries, would we expect to detect a strongly beneficial allele that reached fixation in only one of these ancestries? But then to address this question, the authors do something quite odd in my view, which is to consider that one step back, populations suddenly have a meaning again, and admixture is not an issue. Specifically, they consider the three ancestries that gave rise to modern Europeans, and imagine those to have persisted long enough for beneficial alleles to ascend from rare to fixation. Yet we know that those ancestries too are mixtures of other ancestries. (Contrary to what is stated in the SOM, forming a blob in a PCA plot is not evidence for them representing an enduring un-admixed population—so do Europeans in the 1000G data for example.) So the whole setup of this study seems inconsistent to me.

In a sense, the question that the authors raise is in my view deeper than what they contend with here: given that mixing of divergent groups was the norm throughout human evolution, likely at rates higher than the time scale it takes for a rare mutation to reach fixation (~50,000 years?), are hard sweeps even the right model for the behavior of strongly beneficial alleles? By using the hard sweep model, could we be missing quasi-Mendelian adaptations?

R1: The authors respond to this point by acknowledging that the three ancestral populations could also be admixed but give arguments to support that there was more opportunity for strongly beneficial alleles to reach fixation in these ancestral populations than in more modern populations. They also cite Lazaridis et al. 2016 and Mathieson et al. 2018 as evidence that Anatolia, WHG and Steppe provide the predominant sources for Eurasian populations included in their study. I find their arguments convincing but the authors should include these citations in line 41 and mention that the ancestral populations could also be admixed. Additionally, the authors could acknowledge that their demographic model might not be accurately capturing the true complexity of the population's demographic history?

We have now added citations to Lazaridis et al. 2016 and Mathieson et al. 2018 to the manuscript, and mention that the ancestral populations could be admixed and thus contain masked sweeps themselves (Discussion, lines 278-287):

"While our analyses point to well-known admixture events during the Holocene as the prime driver of the diluted sweep signals observed in modern European genomes, it is possible that the three populations directly ancestral to present-day Europeans (i.e. Mesolithic hunter-gatherers, Anatolian Neolithic farmers, and pastoralists from the Pontic-Caspian steppe) were also admixed to some degree (Marchi et al. 2022). However, the much stronger genetic differentiation observed between the three ancestral populations relative to the later Holocene European groups (Lazaridis et al. 2016) suggests that potential admixture events involving the three ancestral lineages were probably less impactful or frequent than subsequent admixture phases in the Holocene (Lazaridis et al. 2016). This implies less perturbation of any underlying sweep signals, although we note that the occurrence of such mixing events would mean that historical hard sweeps were even more frequent than identified in our study."

We also further acknowledge that our demographic model might not be accurately capturing the true complexity of the population's demographic history (page 6, lines 166-169):

"Although our simulations might not capture the full complexity of West Eurasian demographic history, the model has sufficient detail to provide general insights into the effect of admixture on signatures of hard sweeps in these lineages."

R3: Instead, this study just kicks the can down the road, by considering DNA samples from the three labels that mixed to form present day Europeans as populations, and looking for sweeps in them instead. So then we have to ask ourselves how errors in the specification of the demographic model for the "ancestral populations" that gave rise to current day Europeans

could lead to false positives in their sweep detection. Given that mis-specification of the demographic history is a huge issue in that regard, some sanity checks are in order. (To be clear, the authors present many analyses of possible artifacts, but none on what seems likely to be the thorniest issue.) Likewise, the lessons of their Figures 4-6 are dependent on the specifics of the simple demographic model they choose, so it would be good to know how much independent support there is for their model and how well it fits data at neutral sites.

R1: I agree with the authors' response that shows that they do account for demographic history misspecification by testing different bottleneck strengths and admixture proportions.

We thank Reviewer 1 for acknowledging the work evaluating and confirming the robustness of our method to bottlenecks and admixture.

R3: In that regard, it seems a weird omission that the authors don't tell us about 56 of the 57 sweep signals not in the MHC, other than that they have higher F_{st} to Yoruba: where are they? are they plausible candidates etc?

R1: In the response to Reviewer 3, the authors write that a large proportion of their 57 sweep regions appear in previous studies of modern European populations, including some well-established genes of strong adaptive evolution such as SLC24A5, MLPH, HLA and others. This is confusing since in line 82 they mention that only two of the 57 SF2 outliers are detected in modern populations and have been reported in previous studies. Then, in line 156 they mention that ~66% of the 57 sweeps appear as partial sweeps in prior studies of positive selection in modern European populations. In their response, the authors write that "our candidate regions are highly consistent with previous literature, although we differ in the interpretation of the spatio-temporal context of selection". What do they mean by this spatio-temporal interpretation? Is it that the sweeps appear as partial and not complete sweeps in modern populations? If sweeps are hard pre and post admixture I don't think they are being misinterpreted. The authors should be clearer on which of their 57 sweep regions overlap with previous literature and clarify what they mean by not agreeing with the "spatio-temporal context of selection" of these previously identified sweeps.

We thank the Reviewer for highlighting the unclear phrasing in line 82. This referred to the fact that only two of the 57 ancient sweeps of our study are significant in one of the modern European populations (CEU, FIN, TSI), when using our method for detecting hard sweeps close to fixation. We have now removed the last part of this sentence referring to these two loci also having been reported in previous selection studies – as this might lead to the incorrect impression that *only* these two loci have been found in previous selection studies. Our main point in this section of the manuscript is the dramatic reduction in fixed hard sweep signals over time. We show which of the 57 sweep regions overlap with previous literature in Table S2.

Further, we confirm that regarding the "spatio-temporal interpretation" we refer to the fact that a relatively large proportion (70%) of our 57 sweeps appear as haplotype-based *partial* sweep signals in previous studies of modern populations, i.e. not fixed sweep signals. As noted

above, haplotype-based signals of selection are typically interpreted as very recent and ongoing selection of variants that have not yet reached fixation (e.g., Günther and Schmid 2011; Ronen et al. 2015; Sabeti 2006; Voight et al. 2006; Johnson and Voight 2018). However, our results suggest that the sweeps are considerably more ancient than expected based on the haplotype-based signal (84% appear older than 30,000 years), and that the selection pressures underlying the sweeps may have eased during the Holocene period and might thus not constitute *recent* selection. We clarify this aspect now in the Discussion (see also our response above):

"Our empirical and simulation results implicate Holocene-era admixture as the primary factor attenuating these historical sweep signals, which has led to them being missed in previous studies, or detected instead as empirical genome-wide outliers of haplotype-based selection statistics (Pickrell et al. 2009; Johnson and Voight 2018). Such haplotype-based outliers are typically interpreted as ongoing partial sweeps resulting from recent selection (Sabeti 2006; Johnson and Voight 2018; Günther and Schmid 2011; Ronen et al. 2015; Voight et al. 2006), as the underlying haplotype patterns decay quickly over time and become largely undetectable for selection starting more than 30,000 years ago in humans (Sabeti 2006; Voight et al. 2006). However, our analyses suggest that most (85%) of the 41 sweeps that overlap with a haplotype-based outlier were already under selection by 30ka, implying that these signals more likely result from admixture-driven dilution of hard sweeps that mostly began before 30,000 years ago."

Regarding the few well-established genes of strong evolution that also appear as outliers in our study, we want to note that two of them appear as partial sweeps in modern data (MLPH (Pickrell et al. 2009), ATXN2 (Mathieson et al. 2015)), one as an outlier in a non-European population (East Asians for EDAR (Mathieson et al. 2015)), and one as a signal of balancing selection (HLA (Bitarello et al. 2018)). Thus, even though they have been reported as being under selection previously, we are the first to report a signal of an ancient hard selective sweep close to fixation for these genes - revealing a much longer evolutionary history.

R3: Finally, the section on the mutational basis of sweeps is missing previous references about what happens when previously deleterious mutations become strongly favored, and how these are not distinguishable from hard sweeps, as well as the interaction with demography, such as Orr & Betancourt 2001 Genetics and Teshima et al. 2006 Genome Research, among others.

R1: The authors added citations to previous literature on selection from the standing variation.

References

- Bitarello, Bárbara D., Cesare de Filippo, João C. Teixeira, Joshua M. Schmidt, Philip Kleinert, Diogo Meyer, and Aida M. Andrés. 2018. "Signatures of long-term balancing selection in human genomes." *Genome Biology and Evolution* 10 (3): 939–55.
- Günther, Torsten, and Karl J. Schmid. 2011. "Improved haplotype-based detection of ongoing selective sweeps towards an application in *Arabidopsis thaliana*." *BMC Research Notes* 4 (July): 232.
- Huber, Christian D., Magnus Nordborg, Joachim Hermisson, and Ines Hellmann. 2014. "Keeping it local: evidence for positive selection in Swedish *Arabidopsis thaliana*." *Molecular Biology and Evolution* 31 (11): 3026–39.
- Johnson, Kelsey Elizabeth, and Benjamin F. Voight. 2018. "Patterns of shared signatures of recent positive selection across human populations." *Nature Ecology & Evolution* 2 (4): 713–20.
- Lazaridis, Iosif, Dani Nadel, Gary Rollefson, Deborah C. Merrett, Nadin Rohland, Swapan Mallick, Daniel Fernandes, et al. 2016. "Genomic insights into the origin of farming in the Ancient Near East." *Nature* 536 (7617): 419–24.
- Marchi, Nina, Laura Winkelbach, Ilektra Schulz, Maxime Bami, Zuzana Hofmanová, Jens Blöcher, Carlos S. Reyna-Blanco, et al. 2022. "The genomic origins of the world's first farmers." *Cell* 185 (11): 1842–59.e18.
- Mathieson, Iain, Iosif Lazaridis, Nadin Rohland, Swapan Mallick, Nick Patterson, Songül Alpaslan Roodenberg, Eadaoin Harney, et al. 2015. "Genome-wide patterns of selection in 230 ancient Eurasians." *Nature* 528 (7583): 499–503.
- Pickrell, Joseph K., Graham Coop, John Novembre, Sridhar Kudaravalli, Jun Z. Li, Devin Absher, Balaji S. Srinivasan, et al. 2009. "Signals of recent positive selection in a worldwide sample of human populations." *Genome Research* 19 (5): 826–37.
- Ronen, Roy, Glenn Tesler, Ali Akbari, Shay Zakov, Noah A. Rosenberg, and Vineet Bafna. 2015. "Predicting carriers of ongoing selective sweeps without knowledge of the favored allele." *PLoS Genetics* 11 (9): e1005527.
- Sabeti PC, Schaffner SF, Fry B, Lohmueller J, Varilly P, Shamovsky O, Palma A, Mikkelsen TS, Altshuler D, Lander ES. 2006. "Positive natural selection in the human lineage." *Science* 312 (5780): 1614-1620.
- Voight, Benjamin F., Sridhar Kudaravalli, Xiaoquan Wen, and Jonathan K. Pritchard. 2006. "A map of recent positive selection in the human genome." *PLoS Biology* 4 (3): e72.
- Zheng, Yichen, and Thomas Wiehe. 2019. "Adaptation in structured populations and fuzzy boundaries between hard and soft sweeps." *PLoS Computational Biology* 15 (11): e1007426.

Decision Letter, second revision:

12th July 2022

Dear Dr. Souilmi,

Thank you for submitting your revised manuscript "Admixture has obscured signals of historical hard sweeps in humans" (NATECOLEVOL-211115191B). It has now been seen again by the original reviewers and their comments are below. The reviewers find that the paper has improved in revision, and therefore we'll be happy in principle to publish it in Nature Ecology & Evolution, pending minor revisions to comply with our editorial and formatting guidelines.

[REDACTED]

Reviewer #1 (Remarks to the Author):

We thank the authors for their thorough responses to our reviews and are satisfied with their answers.

Our ref: NATECOLEVOL-211115191B

24th August 2022

Dear Dr. Tobler,

34Thank you for your patience as we've prepared the guidelines for final submission of your Nature Ecology & Evolution manuscript, "Admixture has obscured signals of historical hard sweeps in humans" (NATECOLEVOL-211115191B). Please carefully follow the step-by-step instructions provided in the attached file, and add a response in each row of the table to indicate the changes that you have made. Please also check and comment on any additional marked-up edits we have proposed within the text. Ensuring that each point is addressed will help to ensure that your revised manuscript can be swiftly handed over to our production team.

****We would like to start working on your revised paper, with all of the requested files and forms, as soon as possible (preferably within two weeks). Please get in contact with us immediately if you anticipate it taking more than two weeks to submit these revised files.****

In recognition of the time and expertise our reviewers provide to Nature Ecology & Evolution's editorial process, we would like to formally acknowledge their contribution to the external peer review of your manuscript entitled "Admixture has obscured signals of historical hard sweeps in humans". For those reviewers who give their assent, we will be publishing their names alongside the published article.

Nature Ecology & Evolution offers a Transparent Peer Review option for new original research manuscripts submitted after December 1st, 2019. As part of this initiative, we encourage our authors to support increased transparency into the peer review process by agreeing to have the reviewer comments, author rebuttal letters, and editorial decision letters published as a Supplementary item. When you submit your final files please clearly state in your cover letter whether or not you would like to participate in this initiative. Please note that failure to state your preference will result in delays in accepting your manuscript for publication.

Cover suggestions

As you prepare your final files we encourage you to consider whether you have any images or illustrations that may be appropriate for use on the cover of Nature Ecology & Evolution.

35If your image is selected, we may also use it on the journal website as a banner image, and may need to make artistic alterations to fit our journal style.

Nature Ecology & Evolution has now transitioned to a unified Rights Collection system which will allow our Author Services team to quickly and easily collect the rights and permissions required to publish your work. Approximately 10 days after your paper is formally accepted, you will receive an email in providing you with a link to complete the grant of rights. If your paper is eligible for Open Access, our Author Services team will also be in touch regarding any additional information that may be required to arrange payment for your article.

Please note that *Nature Ecology & Evolution* is a Transformative Journal (TJ). Authors may publish their research with us through the traditional subscription access route or make their paper immediately open access through payment of an article-processing charge (APC). Authors will not be required to make a final decision about access to their article until it has been accepted. [Find out more about Transformative Journals](https://www.springernature.com/gp/open-research/transformative-journals)

Authors may need to take specific actions to achieve [compliance with funder and institutional open access mandates](https://www.springernature.com/gp/open-research/funding/policy-compliance-faqs). If your research is supported by a funder that requires immediate open access (e.g. according to [Plan S principles](https://www.springernature.com/gp/open-research/plan-s-compliance)) then you should select the gold OA route, and we will direct you to the compliant route where possible. For authors selecting the subscription publication route, the journal's standard licensing terms will need to be accepted, including [those licensing terms](https://www.nature.com/nature-portfolio/editorial-policies/self-archiving-and-license-to-publish) will supersede any other terms that the author or any third party may assert apply to any version of the manuscript.

Please use the following link for uploading these materials:
[REDACTED]

[REDACTED]

Reviewer #1:

Remarks to the Author:

We thank the authors for their thorough responses to our reviews and are satisfied with their answers.

Author rebuttal, second revision:

Reviewer #1 (Remarks to the Author):

We thank the authors for their thorough responses to our reviews and are satisfied with their answers.

We thank Reviewer 1 for their final evaluation.

Final Decision Letter:

16th September 2022

Dear Dr Tobler,

We are pleased to inform you that your Article entitled "Admixture has obscured signals of historical hard sweeps in humans", has now been accepted for publication in Nature Ecology & Evolution.

Over the next few weeks, your paper will be copyedited to ensure that it conforms to Nature Ecology and Evolution style. Once your paper is typeset, you will receive an email with a link to choose the appropriate publishing options for your paper and our Author Services team will be in touch regarding any additional information that may be required

37You will not receive your proofs until the publishing agreement has been received through our system

Due to the importance of these deadlines, we ask you please us know now whether you will be difficult to contact over the next month. If this is the case, we ask you provide us with the contact information (email, phone and fax) of someone who will be able to check the proofs on your behalf, and who will be available to address any last-minute problems . Once your paper has been scheduled for online publication, the Nature press office will be in touch to confirm the details.

Acceptance of your manuscript is conditional on all authors' agreement with our publication policies (see www.nature.com/authors/policies/index.html). In particular your manuscript must not be published elsewhere and there must be no announcement of the work to any media outlet until the publication date (the day on which it is uploaded onto our web site).

Please note that *Nature Ecology & Evolution* is a Transformative Journal (TJ). Authors may publish their research with us through the traditional subscription access route or make their paper immediately open access through payment of an article-processing charge (APC). Authors will not be required to make a final decision about access to their article until it has been accepted. [Find out more about Transformative Journals](https://www.springernature.com/gp/open-research/transformative-journals)

Authors may need to take specific actions to achieve [compliance](https://www.springernature.com/gp/open-research/funding/policy-compliance-faqs) with funder and institutional open access mandates. If your research is supported by a funder that requires immediate open access (e.g. according to [Plan S principles](https://www.springernature.com/gp/open-research/plan-s-compliance)) then you should select the gold OA route, and we will direct you to the compliant route where possible. For authors selecting the subscription publication route, the journal's standard licensing terms will need to be accepted, including [those licensing terms](https://www.nature.com/nature-portfolio/editorial-policies/self-archiving-and-license-to-publish) will supersede any other terms that the author or any third party may assert apply to any version of the manuscript.

We welcome the submission of potential cover material (including a short caption of around 40 words) related to your manuscript; suggestions should be sent to Nature Ecology & Evolution as electronic files (the image should be 300 dpi at 210 x 297 mm in either TIFF or JPEG format). Please note that such pictures should be selected more for their aesthetic appeal than for their scientific content, and that colour images work better than black and white or grayscale images. Please do not try to design a cover with the Nature Ecology & Evolution logo etc., and please do not submit composites of images related to your work. I am sure you will understand that we cannot make any promise as to whether any of your suggestions might be selected for the cover of the journal.

You can generate the link yourself when you receive your article DOI by entering it here: <http://authors.springernature.com/share>.

[REDACTED]

P.S. Click on the following link if you would like to recommend Nature Ecology & Evolution to your librarian <http://www.nature.com/subscriptions/recommend.html#forms>

** Visit the Springer Nature Editorial and Publishing website at http://editorial-jobs.springernature.com?utm_source=ejp_NEcoE_email&utm_medium=ejp_NEcoE_email&utm_campaign=ejp_NEcoE for more information about our career opportunities. If you have any questions please click [here](mailto:editorial.publishing.jobs@springernature.com). **